# The oncolytic virus Delta-24-RGD elicits an antitumor effect in pediatric glioma and DIPG mouse models

Naiara Martínez-Vélez[1,2,3], Marc Garcia-Moure[1,2,3], Miguel Marigil[1,2,3], Marisol González-Huarriz[1,2,3], Montse Puigdelloses[1,2,4], Jaime Gallego Pérez-Larraya [1,2,4], Marta Zalacaín[1,2,3], Lucía Marrodán[1,2,3], Maider Varela-Guruceaga[1,2,3], Virginia Laspidea[1,2,3], Jose Javier Aristu[1,5], Luis Isaac Ramos[1,5], Sonia Tejada-Solís[1,6], Ricardo Díez-Valle[1,6], Chris Jones[7,8], Alan Mackay[7,8], Jose A. Martínez-Climent[1,9], Maria Jose García-Barchino[1,9], Eric Raabe [10,11], Michelle Monje [12], Oren J. Becher[13], Marie Pierre Junier[14], Elias A. El-Habr[14], Herve Chneiweiss [14], Guillermo Aldave[15], Hong Jiang [16], Juan Fueyo[16,17], Ana Patiño-García [1,2,3], Candelaria Gomez-Manzano[16] & Marta M. Alonso [1,2,3]

Pediatric high-grade glioma (pHGG) and diffuse intrinsic pontine gliomas (DIPGs) are aggressive pediatric brain tumors in desperate need of a curative treatment. Oncolytic virotherapy is emerging as a solid therapeutic approach. Delta-24-RGD is a replication competent adenovirus engineered to replicate in tumor cells with an aberrant RB pathway. This virus has proven to be safe and effective in adult gliomas. Here we report that the administration of Delta-24-RGD is safe in mice and results in a significant increase in survival in immunodeficient and immunocompetent models of pHGG and DIPGs. Our results show that the Delta-24-RGD antiglioma effect is mediated by the oncolytic effect and the immune response elicited against the tumor. Altogether, our data highlight the potential of this virus as treatment for patients with these tumors. Of clinical significance, these data have led to the start of a phase I/II clinical trial at our institution for newly diagnosed DIPG (NCT03178032).

[1] Health Research Institute of Navarra (IDISNA), Pamplona, Navarra, Spain. [2] Program of Solid Tumors, Center for the Applied Medical Research (CIMA), University of Navarra, Navarra, Pamplona, Spain. [3] Department of Pediatrics, Clínica Universidad de Navarra, Pamplona, Spain. [4] Department of Neurology, Clínica Universidad de Navarra, Pamplona, Spain. [5] Department of Radiation Oncology, Clínica Universidad de Navarra, Pamplona, Spain. [6] Department of Neurosurgery, Clínica Universidad de Navarra, Pamplona, Spain. [7] Division of Molecular Pathology, The Institute of Cancer Research, 15 Cotswold Road, Sutton, London, Surrey SM2 5NG, UK. [8] Division of Cancer Therapeutics, The Institute of Cancer Research, 15 Cotswold Road, Sutton, London, Surrey SM2 5NG, UK. [9] Division of Hematopoietic Tumors, Center for the Applied Medical Research (CIMA), University of Navarra, CIBERONC, Pamplona, Pamplona, Navarra, Spain. [10] Department of Pathology, The Johns Hopkins University School of Medicine, Baltimore, MD, USA. [11] Division of Pediatric Oncology, The Johns Hopkins University School of Medicine, Baltimore, MD, USA. [12] Department of Neurology, Stanford University School of Medicine, Stanford, CA, USA. [13] Department of Pediatrics, Northwestern University and Division of Pediatric Hematology-Oncology and Stem Cell Transplant, Ann & Robert H. Lurie Children's Hospital, Chicago, IL, USA. [14] CNRS UMR8246, Inserm U1130, Neuroscience Paris Seine - IBPS, Sorbonne Universities, Paris, France. [15] Division of Pediatric Neurosurgery, Department of Surgery, Texas Children's Hospital, Baylor College of Medicine, Houston, TX, USA. [16] Department of NeuroOncology, The University of Texas MD Anderson Cancer Center, Houston, TX, USA. [17] Department of Neurosurgery, The University of Texas MD Anderson Cancer Center, Houston, TX, USA. Correspondence and requests for materials should be addressed to C.G.-M. (email: cmanzano@mdanderson.org) or to M.M.A. (email: mmalonso@unav.es)

Pediatric high-grade glioma (pHGG), including diffuse intrinsic pontine glioma (DIPG), are aggressive solid tumors that develop during childhood. pHGG was traditionally considered similar to adult high-grade gliomas; however, over the last decade, an emerging plethora of genomic data has changed our understanding of both pHGG and DIPG and demonstrated that these cancers are indeed very different entities from their adult counterparts[1–3]. The knowledge acquired from genomic data has supported the 2016 WHO reclassification of central nervous system (CNS) tumors in which DIPG is included in a new category named diffuse midline glioma (DMG) H3-K27M-mutant[4].

Regarding their epidemiology, these two malignancies account for ~ 8–12% of all CNS tumors that arise in children. Because the incidence of this disease is ~ 85 per 100,000, similar to most pediatric tumors, pHGG and DIPG are considered orphan diseases[5]. The incidence peak of pHGG and DIPG appears in patients aged between 6 and 9 years[6]. The standard treatment for pHGG is maximal surgical resection, followed by radiotherapy and/or temozolomide cycles[7]. Unfortunately, surgery is not a therapeutic option for DIPG, and the standard of care is radiotherapy (RT). RT improves the quality of life and survival of these children; however, RT is not curative[8]. Despite the significant number of clinical trials investigating pHGG and DIPG, patient outcomes have continued to be dismal over the last four decades[9,10]. The implementation of alternative therapies that challenge the actual therapeutic paradigm could provide a breakthrough in the treatment of these tumors.

Virotherapy is emerging as an alternative treatment for cancer. In fact, the recent approval of T-VEC (Talimogene laherparepvec), which is an oncolytic herpes virus, by the FDA for metastatic melanoma has enabled the possibility to use other oncolytic viruses as standard treatment for cancer[11].

Delta-24-RGD is a replicative oncolytic adenovirus specifically modified to destroy cancer cells[12,13]. A genetic modification in the E1A viral gene, which is a 24-base pair deletion responsible for pRB protein binding, confers this adenovirus with selectivity to cancer cells. The addition of an RGD sequence in the viral fiber improves its infectivity, allowing the virus to interact with $\alpha v \beta_3$ and $\alpha v \beta_5$ integrins that are overexpressed in a wide range of tumors, including glioma cells. In fact, several clinical trials involving adult glioma patients have shown that the intratumoral administration of Delta-24-RGD (DNX-2401 in the clinic) is safe and provides a therapeutic advantage in many patients[14]. Recent studies have demonstrated that the antitumor effect displayed by oncolytic adenoviruses is not only related to their intrinsic oncolytic effect but also to the triggering of an immune response. Tumor infection elicits specific antitumor immunity in adult mouse glioma models. Importantly, these results have been confirmed in patients, and histological analyses of tumor resections after Delta-24-RGD administration have shown an increased number of infiltrated immune populations in the tumors, suggesting that an immune-mediated antiglioma response occurred[14]. These results encourage the translation of Delta-24-RGD as a feasible treatment for pediatric brain tumors.

In this study, we evaluate the antiglioma effect of Delta-24-RGD in pHGG and DIPG models. Our data show that Delta-24-RGD results in a significant antitumor effect in vitro in a panel of cell lines and in vivo in pHGG and DIPG orthotopic immunosuppressed and immunocompetent models. Our results show that in addition to the oncolytic effect, Delta-24-RGD administration triggers an antitumor immune response. These promising preclinical results paved the way for a phase I/II clinical trial investigating DNX-2401 (NCT03178032) for the treatment of newly diagnosed DIPGs at our institution[15].

## Results

### Delta-24-RGD determinants of infection and replication in pHGG/DIPG

To evaluate whether pHGG and DMG/DIPG are susceptible to viral infection, using a cohort of 220 patient samples of pHGG and DMG/DIPG previously published[1], we performed "in silico analyses" of the expression of the following main receptors that use Delta-24-RGD to enter cells: Coxsackie adenovirus receptor (CAR; CXADR), integrinαVβ3 (ITGA3), integrinαVβ5 (ITGA5), and integrinαVβ (ITGAV). Our analyses revealed that independently of the molecular group to which the patients were assigned, all patients expressed significant amounts of these genes (Fig. 1a and Supplementary Fig. 1a). These data indicate that these tumors are potentially susceptible to Delta-24-RGD infection. Then, as Delta-24-RGD replication is dependent on an aberrant RB pathway, we analyzed the expression and amplifications/deletions of the genes involved in the RB pathway, specifically cdk4/6, cyclin D1, RB, and E2F1. We found that different RB pathway profiles are related to the different molecular subgroups (H3.3G34R, H3.3K27M, H3.1K27M, IDH1, BRAF_PXA, and HGG_GBM_wt) (Fig. 1b, Supplementary Fig. S1b). cdk4/6 had a modest rate in all subgroups; meanwhile, CCND1 amplification was only observed in the H3.3K27M molecular subgroup. RB deletions were only present in the IDH1 group. However, CDKN2A deletions were found in almost all subgroups (Fig. 1b), and this gene encodes p16INK4a and p14arf. As a result of all these deletions, we observed increased E2F1 expression, which, in turn, activates the viral transcriptional program and finally its replication (Supplementary Fig. 1c). In summary, these results highlight the suitability of Delta-24-RGD as a possible therapy for pHGG and DMG, including DIPGs.

### Delta-24-RGD antiglioma effect in human pHGG and DIPG cell lines

The analysis of viral receptor ($\alpha v \beta_3$, $\alpha v \beta_5$ integrin, and CAR) expression in the membrane of a panel of pHGGs and DIPGs revealed that all display these receptors in their membrane, suggesting that they could be infected by an adenovirus. Interestingly, the CAR receptor was abundantly expressed in all cell lines assessed (ranging from 40 to 90%) (Fig. 2a). These results are consistent with the expression data observed in the patients (Fig. 1). Then, we tested the capacity of Delta-24-RGD to infect pHGG (CHLA-03-AA, CHLA-200, PBT-24, SJ-GBM2, and SF188) and DIPG (TP54, TP80, TP83, TP84, and SU-DIPG IV) cell lines. We infected these cell lines with a replication-deficient Ad5-GFP-RGD adenovirus at 10 and 100 MOIs and quantified the % of GFP-expressing cells by flow cytometry. We observed that at the 10 MOI, > 50% of the cells were infected, and 100% infection of the cell lines was achieved at 100 MOI (Fig. 2b, Supplementary Fig. 2a). In general, the pHGG cell lines were more easily infected than the DIPG cell lines.

Following the infection of the indicated cell lines with increasing amounts of Delta-24-RGD, E1A, and fiber, which are an early protein and master regulator of the viral cycle and a late protein, respectively, showed a robust expression. The expression of these proteins was patent even at the smallest dose of five MOI (Fig. 2c and Supplementary Fig. 2b). These results suggest that Delta-24-RGD is able to undergo a viable viral cycle in pHGG and DIPG cells. Furthermore, Delta-24-RGD efficiently replicated with at least an increment of two logarithms in viral progeny compared with the input virus ($10^6$ pfu, dashed line) in all cell lines tested after 72 h of infection (Fig. 2d). Overall, the virus showed a better replication in the pHGG cell lines when compared with the DIPG cell lines ($P = 0.03$; Supplementary Fig. 2c).

Subsequently, we evaluated the antitumor effect of Delta-24-RGD in the pHGG ($N = 5$) and DIPG ($N = 7$) cell lines.

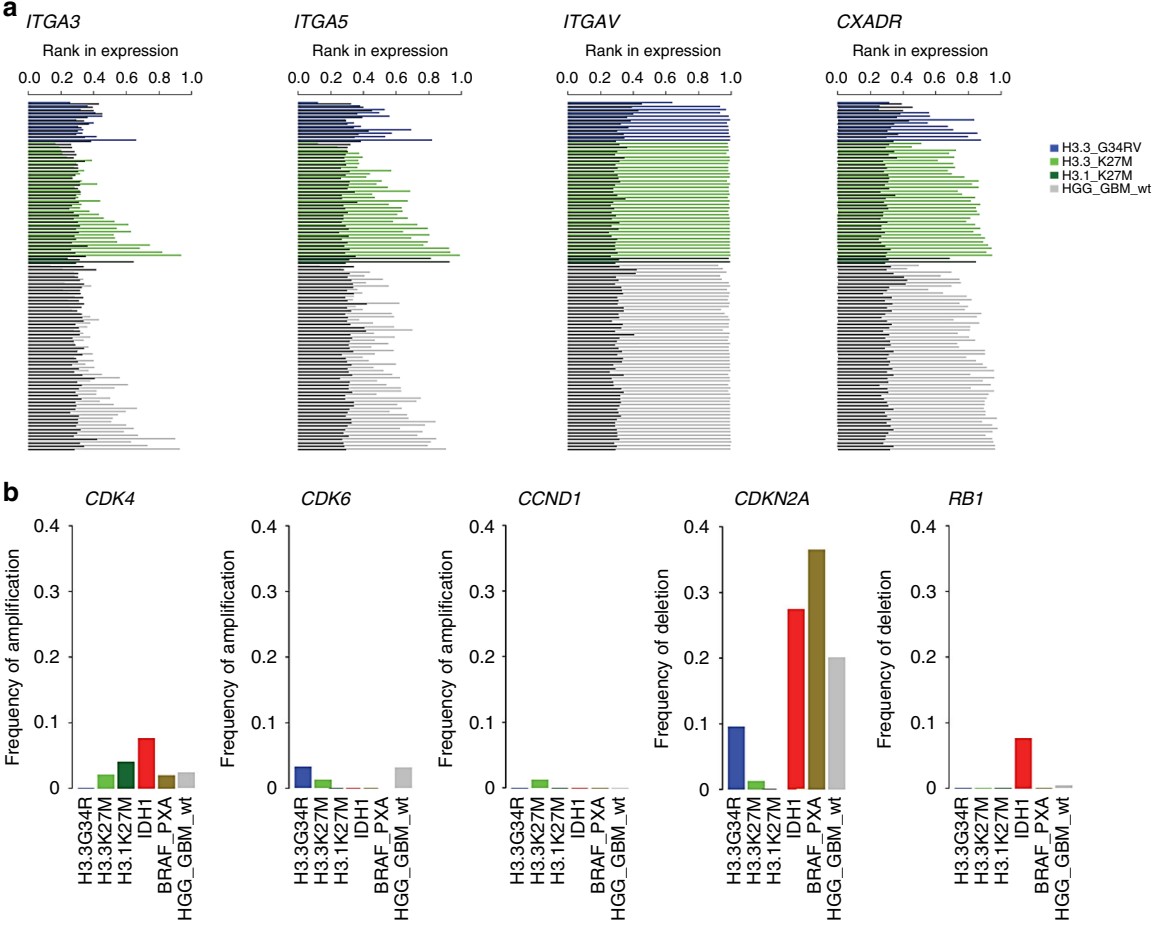

**Fig. 1** Delta-24-RGD determinants of replication and infection in pHGG and DIPG. **a** Assessment of integrin and CAR expression in pHGG and DIPGs. **b** Characterization of amplifications and deletions of RB pathway genes in pHGG and DIPGs. Source data are provided as a Source Data file

| Table 1 Median-effect doses (IC$_{50}$) of Delta-24-RGD in pHGG and DIPG cell lines | | |
|---|---|---|
| | **Cell lines** | **IC$_{50}$ (MOIs ± SD)** |
| pHGG | CHLA-03-AA | 10.22 ± 3.1 |
| | PBT-24 | 0.8 ± 0.06 |
| | SF188 | 9.6 ± 0.5 |
| | SJ-GBM2 | 32 ± 7.1 |
| | CHLA-200 | 5 ± 0.06 |
| DIPG | TP54 | 9.6 ± 3.1 |
| | DIPG IV | 3.6 ± 1.8 |
| | TP80 | 39.7 ± 10.9 |
| | TP83 | 51.9 ± 13.7 |
| | TP84 | 68 ± 4.1 |
| | JHH-DIPG1 | 26.2 ± 1.5 |
| | JHH-DIPG16A | 9.6 ± 5.9 |

The IC$_{50}$ value (MOIs) is the median-effect dose (dose affecting 50% of the cells, i.e., 50% survival). IC$_{50}$ = 50% inhibitory concentration; MOI=multiplicity of infection

The Delta-24-RGD treatment resulted in a significant antitumor effect in all cell lines tested. The IC$_{50}$ ranged between 3.6 and 68 MOIs in the DIPG cell lines and 0.8 and 32 MOIs in the pHGG cell lines (Fig. 2e, Supplementary Fig. 3 and Table 1). PBT-24, which is a pHGG cell line in which Delta-24-RGD presented the best infectivity and replication capacity, also displayed the lowest IC$_{50}$. In summary, these results demonstrate that Delta-24-RGD is able to infect, replicate and exert a significant cytotoxic effect in all pHGG and DIPG cell lines tested.

**Delta-24-RGD extends survival in pHGG and DIPGs orthotopic models**. Given the encouraging in vitro results, we evaluated the antitumor effect of Delta-24-RGD in vivo in two orthotopic models of DIPG (TP80 and TP54) and two models of pHGG (CHLA-03-AA and PBT-24). A single dose (10$^8$ pfu per animal) of Delta-24-RGD was intratumorally injected on day 3 after cell implantation. The mice were monitored during the experiment until physical decline, such as weight loss or ataxia, was observed. The survival analyses showed that Delta-24-RGD was able to significantly increase ($P = 0.024$, Log-rank test) the overall survival of mice bearing TP80 with a median survival of 217 days. The Delta-24-RGD administration increased survival by an average of 40 days ($P = 0.024$, Log-rank test), and the treatment resulted in 44% of long-term survivors in the treated group (Fig. 3a and Supplementary Fig. 4a). Mice bearing the TP54 cell line in the pons exhibited a median survival of 83.5 days, whereas the mice treated with Delta-24-RGD exhibited a median survival of 95.5 days ($P = 0.04$, Log-rank test) (Fig. 3b and Supplementary Fig. 4a). The median survival of the mice bearing CHLA-03-AA was 46 days, whereas the Delta-24-RGD-treated mice had an increased overall survival by 53 days, resulting in 36% of long-term survivors free of disease ($P < 0.0001$, Log-rank test) (Fig. 3c and Supplementary Fig. 4a). Finally, compared with the mice bearing the PBT-24 control (survival of 75 days), the treated mice displayed a median survival of 100.5 days (Log-rank test; $P = 0.0013$), and there were 33% of long-term survivors in the treatment group (Fig. 3d and Supplementary Fig. 4a).

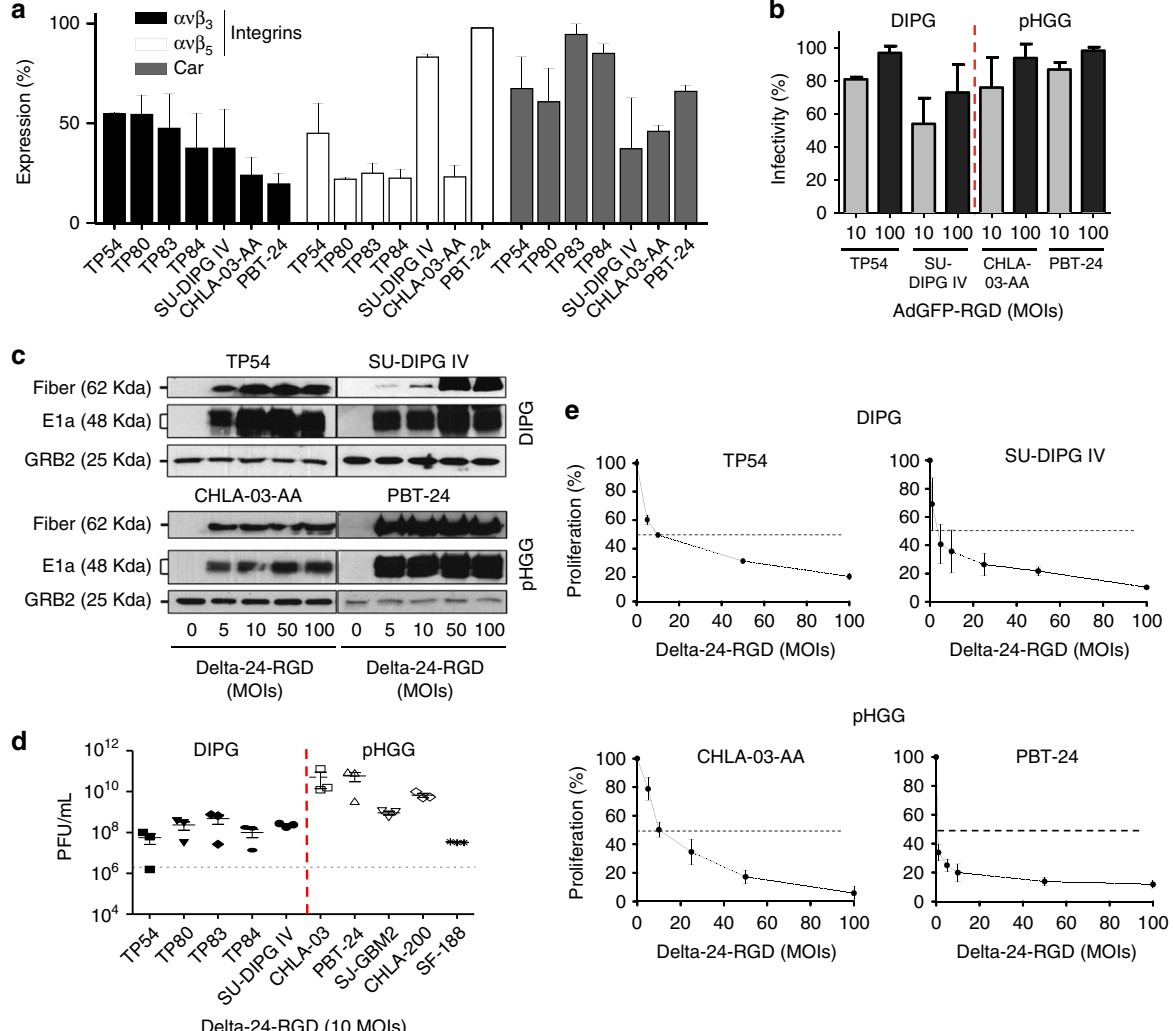

**Fig. 2** Delta-24-RGD exerts a potent oncolytic effect in DIPG and pHGG cell lines. **a** Flow cytometry analyses of CAR and integrin expression. DIPG and pHGG cell lines were incubated with fluorescent antibodies against $\alpha_v\beta_3$ and $\alpha_v\beta_5$ integrins and CAR. The data are shown as the relative percentage (mean ± SD) of positive expression scored among 10,000 cells. **b** Assessment of infectivity in DIPG and pHGG cell lines. The indicated cell lines were infected with a replication-deficient construct expressing a modified fiber knob (AdGFP-RGD). The data are shown as the relative percentage (mean ± SD) of GFP-positive cells scored among 10,000 cells per treatment group. **c** Assessment of viral protein expression in pHGG and DIPG cell lines infected with Delta-24-RGD by western blotting. One representative blot is shown of three independent experiments. **d** Quantification of Delta-24-RGD replication in the indicated cell lines. Viral titers were determined three days after infection at an MOI of 10 ($10^6$ pfu/ml) by an anti-hexon staining-based method in 293 cells and expressed as plaque-forming units (pfu) per milliliter. The dashed line indicates the input virus. The data are shown as the mean ± SD of three independent experiments. **e** Cell proliferation analyses of Delta-24-RGD-infected DIPG and pHGG cell lines. Cell viability was assessed using MTS assays 5 days after infection. The data are shown as the percentage (mean ± SD of three independent experiments) of cells alive after infection with Delta-24-RGD at the indicated multiplicities of infection (MOIs) relative to the non-infected cells (control, equal to 100%). Source data are provided as a Source Data file

The pathological analyses of the tumors showed the highly infiltrative nature of the TP80 cell lines recapitulating the invasive phenotype displayed by midline gliomas. The hematoxylin/eosin staining revealed that the tumor cells infiltrated the whole-brain parenchyma, including other structures, such as the cerebellum (Fig. 3e). TP54, CHLA-03-AA, and PBT-24 are very proliferative tumors with defined borders. The treatment with Delta-24-RGD resulted in smaller tumors and long-term survivors free of disease (Fig. 3f). E1A and hexon expression was detected in the brains of mice treated with the virus (Fig. 3g and Supplementary Fig. 4b, c). The long-term free of disease survivors did not display the expression of viral proteins.

Next, we assessed the antitumor effect of the viral treatment in already established tumor tumors generated by the CHLA-03-AA and PTB-24 (15 days after cell injection; Supplementary Fig. 4d).

The median survival of the mice bearing CHLA-03-AA was 40 days, whereas the Delta-24-RGD-treated mice had an increased median survival of 14.5 days, resulting in 33% of long-term survivors free of disease (Log-rank test; $P = 0.005$) (Fig. 3h). Mice bearing PBT-24 cell line and treated with Delta-24-RGD also displayed a significant increase in overall survival (82 days), compared with the mice bearing the PBT-24 control (76 days; $P = 0.04$, Log-rank test) (Fig. 3i).

**Delta-24-RGD antitumor effects in murine DIPG cell lines.** Clinical trials investigating oncolytic viruses have uncovered that the efficacy of these viruses is partially owed to their potential to awaken the immune system[14,16]. Therefore, our following step was to assess the immune response to Delta-24-RGD treatment in

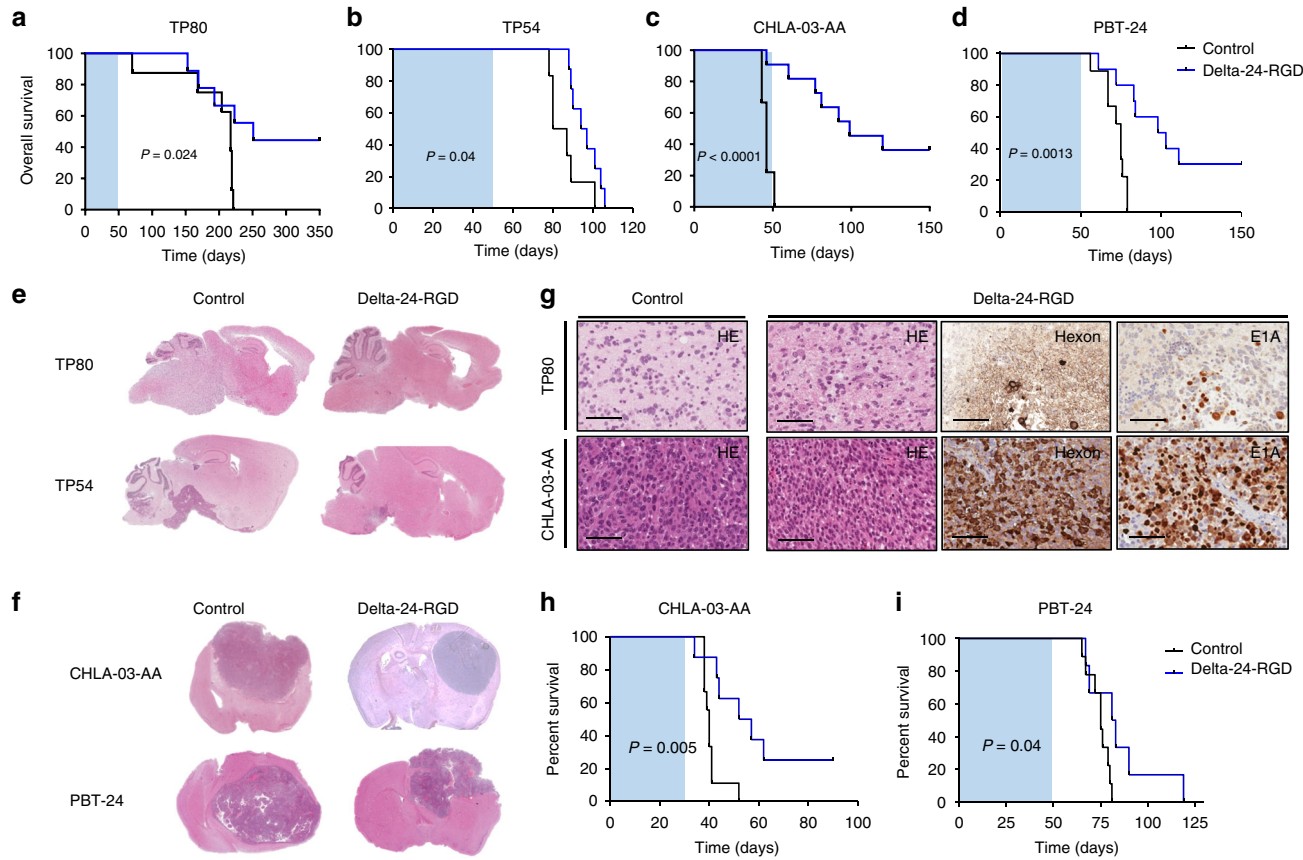

**Fig. 3** Delta-24-RGD increases overall survival in DIPG and pHGG models. Tumors were developed by orthotopic intracranial injection of 500,000 cells in female nude mice. **a** Kaplan–Meier survival curves of Delta-24-RGD ($10^8$ pfu)- and control (PBS)- treated athymic mice bearing **a** TP80 (Control $N = 8$, Delta-24-RGD-treated $N = 9$), **b** TP54 (Control $N = 8$, Delta-24-RGD-treated $N = 9$), **c** CHLA-03-AA (Control $N = 9$, Delta-24-RGD-treated $N = 11$), and **d** PBT-24 (Control $N = 9$, Delta-24-RGD-treated $N = 10$). Animals were treated 3 days post-tumor cell injection. The shaded area represents a 50-day interval from the time of cell implantation. Log-rank test was used as statistical analyses. **e** Representative images of the histopathological analyses of TP80 and TP54 (H&E) of longitudinal slides of control (left images) and Delta-24-RGD-treated tumors (right image). **f** Representative images of the histopathological analyses of CHLA-03-AA and PBT-24 (H&E) (magnification ×1) of control (left image) and Delta-24-RGD-treated (right images). Mice were treated with either PBS (control) or Delta-24-RGD 3 days post injection. For comparison studies, analyzed brains are derived from mice that died at a similar time point in both groups: CHLA-03-AA = 50 ± 10 days; PBT-24 = 60 ± 5 days; TP80 = 200 ± 5 days; TP54 = 90 ± 5 days. **g** Representative images of H&E, hexon, and E1A immunostaining of TP80 or CHLA-03-AA tumors, non-treated, or treated with Delta-24-RGD (scale bars, 100 μm). Images correspond to the brains shown in **e**, **f**. **h**, **i** Kaplan–Meier survival curves of established tumors **h** CHLA-03-AA (Control $N = 10$, Delta-24-RGD-treated $N = 10$) and **i** PBT-24 (Control $N = 8$, Delta-24-RGD-treated $N = 9$) treated with Delta-24-RGD 15 days post cell implantation. The shaded area represents a 50-day interval from the time of cell implantation. Source data are provided as a Source Data file

an immunocompetent model. Thus, we used XFM and NP53 murine cell lines derived from tumors arising in the pons of genetically modified mice[17,18]. The in vitro analyses showed that these cell lines were highly susceptible to adenoviral infection, and at 10 MOI, ~ 100% of the cells were infected (Fig. 4a). One caveat to studies of the immune response to adenoviral infection and replication in immunocompetent mice models is that adenoviral replication (serotype 5) is hindered in murine cell lines[19]. First, we assessed the expression of fiber protein in previously infected NP53 and XFM cell lines. Interestingly, we observed the expression of this late viral protein, suggesting that viral replication might occur in these cells (Fig. 4b). Surprisingly, NP53 and XFM were semi-permissive to viral replication as shown by the viral titers in these cells, which were albeit lower than those in their human counterparts. We observed that the virus replication was higher in NP53 or XFM infected with 300 MOIs (two-tailed Student $t$ test; $P = 0.07$ and $P = 0.005$, respectively) when compared with the initial inoculum (Fig. 4c). We also observed the assembly of virions in the cell nucleus (Fig. 4d). Finally, the cytotoxicity analyses revealed that Delta-24-RGD was able to kill

NP53 and XFM with IC$_{50}$s of 104 and 36 MOIs, respectively (Fig. 4e).

**Delta-24-RGD administration in murine DIPG-immunocompetent models.** We have clinical evidence of the safety of Delta-24-RGD in supratentorial tumors[14]. However, Delta-24-RGD administration into the pons could lead to inflammation and, if uncontrolled, fatalities specifically in DIPG patients. Therefore, we performed a dose escalation study to confirm the lack of toxicity in this model. Previously, we had performed kinetics studies to evaluate the most suitable number of cells for these experiments (Supplementary Fig. 5a). Mice bearing DIPG cells in the pons were intratumorally treated with virus at doses ranging from $10^6$ to $10^8$ pfu in increments of a half logarithm and followed for 15 days to assess toxicity. No dose analyzed caused a lethal reaction (Supplementary Table 2). In addition, we evaluated one injection versus three viral injections every other day at the maximum dose ($10^8$ pfu). Again, under this schedule, we did not observe any toxicity (Supplementary

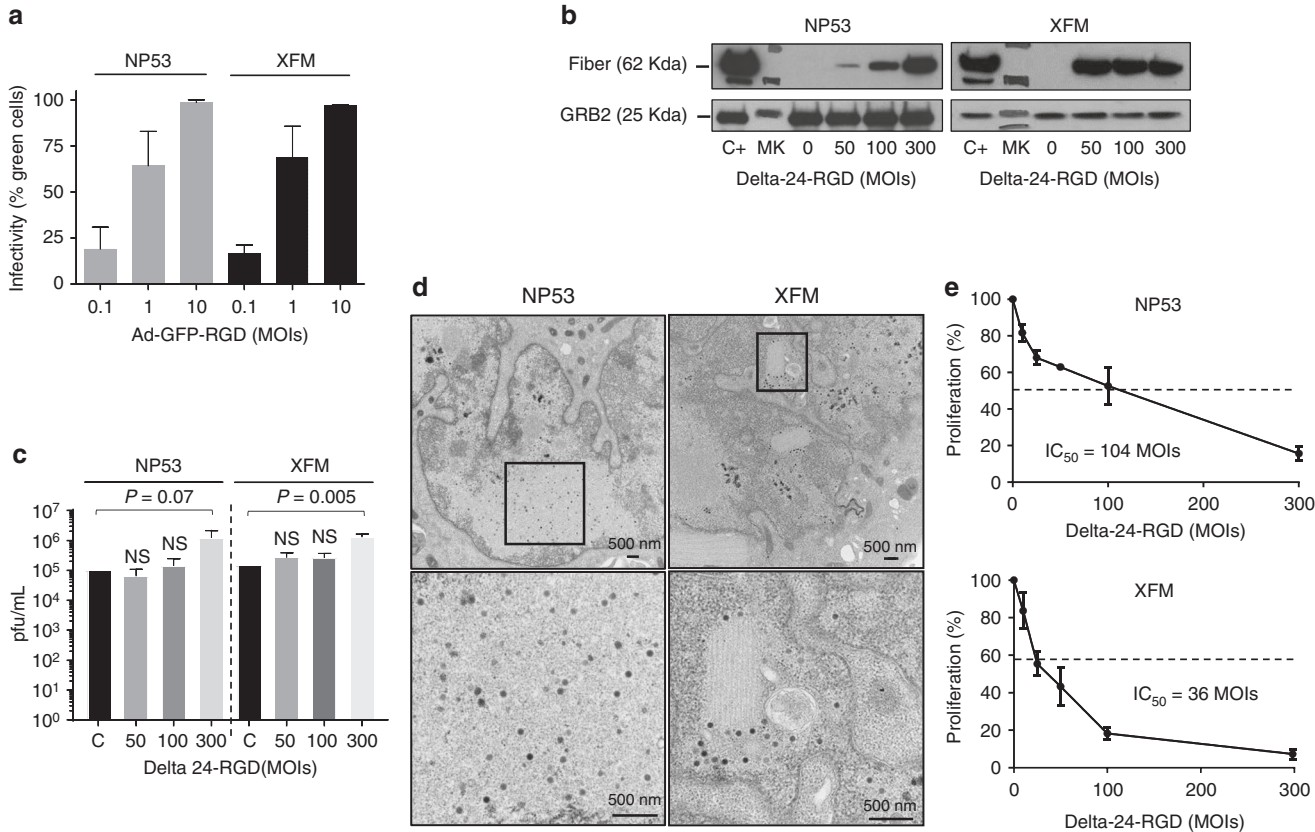

**Fig. 4** Delta-24-RGD exerts a potent oncolytic effect in DIPG murine cell lines. **a** Flow cytometry analyses of infectivity in DIPG murine cell lines. The indicated cell lines were infected at 0.1, 1, and 10 MOIs with a replication-deficient construct expressing a modified fiber knob (AdGFP-RGD). The data are shown as the relative percentage (mean ± SD) of GFP-positive cells scored among 10,000 cells per treatment group. **b** Western blot analysis of late protein fiber expressed in murine cell lines 42 h after infection at different viral doses and a human positive control (C+). MK = Marker. A representative blot is shown of three independent experiments. **c** Quantification of Delta-24-RGD replication in the indicated cell lines. Viral titers were determined 3 days after infection at MOIs of 50, 100, and 300 and control (C) that corresponds to infected cells with 300 MOIs recollected at 16 h by an anti-hexon staining-based method in 293 cells and expressed as plaque-forming units (pfu) per milliliter. The data are shown as the mean ± SD of three independent experiments, two-tailed Student t test analysis was used for comparison. **d** Representative electronic microscopy images showing viral progeny 72 h after viral infection of the NP53 (300 MOIs) or XFM (200 MOIs). **e** Cell proliferation analyses of Delta-24-RGD-infected DIPG murine cell lines. Cell viability was assessed using an MTS assay 5 days after infection. The data are shown as the percentage (mean ± SD of three independent experiments) of cells alive after infection with Delta-24-RGD at the indicated multiplicities of infection (MOIs) relative to the non-infected cells (control, equal to 100%). Source data are provided as a Source Data file

Table 3). As weight loss is a sign of toxicity, we monitored this parameter for 15 days after the virus injection. The animals did not display a significant weight variation related with PBS or Delta-24-RGD administration (Supplementary Fig. 5b). The histology analyses of the DIPG tumors in vivo in mice bearing either NP53 or XFM cells showed that after administration of Delta-24-RGD E1A could be detected in the tumor in both models (Supplementary Fig. 5c). In addition, administration of Delta-24-RGD significantly increased (two-tailed Student t test; $P < 0.0001$) the number of CD3-positive cells infiltrating the tumor mass compared with that in their counterpart controls (Fig. 5a, b and Supplementary Fig. 6a). Moreover, higher CD3+ levels corresponded to higher numbers of CD4- and CD8-positive cells (two-tailed Student t test; $P = 0.0002$, $P = 0.0005$, respectively) and a decrease in FoxP3+/CD4+, which was not significant in the NP53-bearing mice (two-tailed Student t test, $P = 0.07$) but was significant in the XFM-bearing animals (two-tailed Student t test $P = 0.03$) (Fig. 5c–e, and Supplementary Fig. 6b, c). The mRNA expression analysis in the tumors treated with Delta-24RGD revealed higher IFN gamma, CD8a, and CD4 mRNA levels compared with those in the PBS-treated tumors (Fig. 5f and Supplementary Fig. 6d). Moreover, splenocytes extracted from

mice treated with the adenovirus were co-cultured with tumor cells. The splenocytes from the Delta-24-RGD-treated NP53 and XFM-bearing mice expressed significantly more IFN gamma than those extracted from the PBS-treated mice (two-tailed Student t-test; $P = 0.001$, $P = 0.0005$, respectively) (Fig. 5g and Supplementary Fig. 6e), suggesting that there is a specific antitumor immune response.

Once we had proof of the capacity of the virus to trigger an antitumor immune response we evaluated the therapeutic efficacy of Delta-24-RGD in immunocompetent models. First, animals bearing NP53 subcutaneous tumors and treated with Delta-24-RGD displayed a significant lower tumor volume than those treated with PBS (Two-tailed Student t test; $P < 0.05$; Fig. 5h). Next, we evaluated the antitumor effect of Delta-24-RGD in orthotopic immunocompetent mice bearing either NP53 or XFM DIPG murine cell lines. The survival analyses showed that Delta-24-RGD was able to significantly increase the overall survival of mice bearing NP53 or XFM (Log-rank Test; $P = 0.01$ and $P = 0.0002$, respectively) (Fig. 5i and Supplementary Fig. 6f). NP53-bearing mice treated with the virus showed a mean survival of 26 days in comparison with 21 days in PBS-treated mice and led to one (10%) long-term survivor mice (Fig. 5i). Most

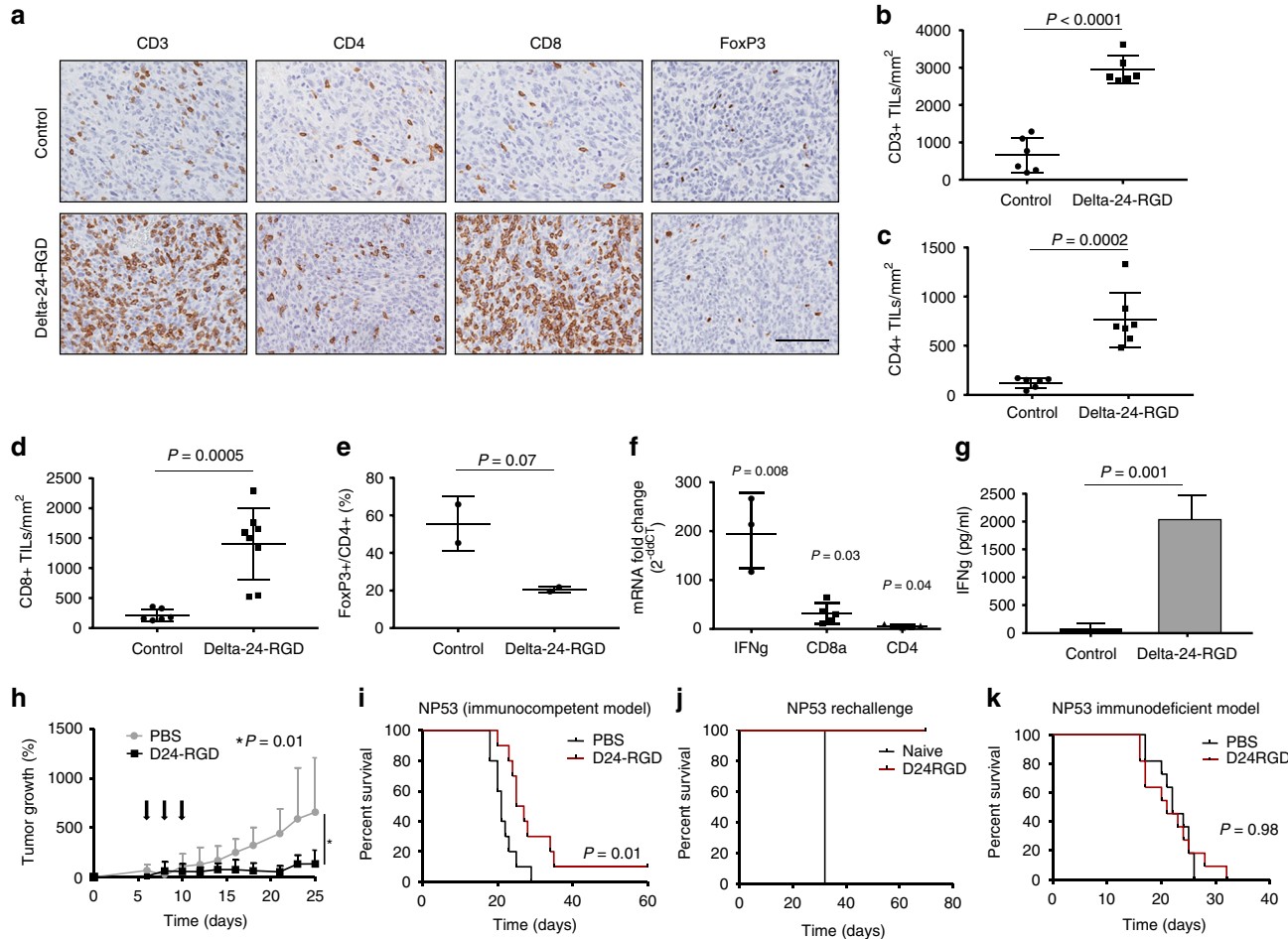

**Fig. 5** Delta-24-RGD effect DIPG immunocompetent models. Brain tumors were developed by intracranial injection of the NP53 cell line, and PBS or Delta-24-RGD were administered intratumorally 3 days after cell implantation and animals were killed at day15 after cell implantation (**a–h**). **a** Representative images (scale bar, 100 μm) of CD3, CD4, CD8, and FoxP3 immunostaining of DIPG tumors from control mice or Delta-24-RGD-treated mice. **b** Quantification of positive CD3+, **c** CD4+, **d** CD8+, and **e** FoxP3/CD4+ cell infiltration per mm² of DIPG tumors. Graph showing the quantification of positive cells infiltrating 15 days after cell implantation per mm² of PBS or Delta-24-RGD-treated tumors (*n* = 3). *P* values were calculated using two-tailed Student *t* test. **f** Quantification of IFN gamma (mean fold change 193.7), CD8a (mean fold change 29.1), and CD4 (mean fold change 2.9) mRNA expression. The data shown represent the mRNA expression in tumors treated with Delta-24-RGD normalized to PBS-tumor mRNA expression (*N* = 3). Two-tailed Student *t* test was used for comparison between control and treated mice. **g** ELISA quantification of IFN gamma production in splenocytes from control and Delta-24-RGD-treated animals co-cultured with tumor cells. *P* values were calculated using two-tailed Student *t* test. **h** NP53 cells (5 × 10⁶) were implanted subcutaneously in mice flank. Seven days later subcutaneous tumors were visible and ranged between 40 and 80 mm³. Mice were randomized in two groups and treated with three administrations of Delta-24-RGD. Tumor volumes were measured every 2–3 days until the end of the experiment (day 25). Tumor volume (*N* = 8) was calculated by the equation $V (mm^3) = \pi/6 \times W^2 \times L$; *W* is tumor width and *L* is tumor length. Graph showing percentage of tumor growth was calculated as $(V - V_0/V_0 \times 100)$ where $V_0$ is the tumor volume present when treatment starts, comparison between groups were performed with two-tailed Student *t* test. **i** Brain tumors were developed by intracranial injection of the NP53 cell line, and PBS or Delta-24-RGD were administered intratumorally 3 days after cell implantation. Kaplan–Meier survival curves of Delta-24-RGD (10⁷ pfu)- and control (PBS)- treated immunocompetent mice (*N* = 10, both groups) bearing intracranial NP53 tumors. *P* value was calculated with the log-rank test. **j** Rechallenge experiment of the long-term survivor from **i**. The long-term survivor from the Delta-24-RGD-treated group was subjected to a rechallenge with NP53 cells and compared with a control untreated mice. **k** Kaplan–Meier survival curves of Delta-24-RGD (10⁷ pfu)- and control (PBS)-treated immunodeficient (athymic nude) mice (*N* = 11; both groups) bearing intracranial NP53 tumors. *P* value was calculated with the log-rank test. Source data are provided as a Source Data file

importantly, Delta-24-RGD treatment of XFM-bearing mice led to 80% of long-term, compared with a median survival of 18 days in PBS-treated XFM-bearing mice (Supplementary Fig. 6f). To demonstrate the generation of an immune memory we performed the reinjection of NP53 and XFM cell lines in the long-term survivors of the previous experiments. Our data showed that animals previously treated with Delta-24-RGD did not develop tumors (Fig. 5j and Supplementary Fig. 6g). To demonstrate that the antitumor response observed in the murine immunocompetent models were owing to an immune response, we performed

similar experiments in athymic nude mice bearing either NP53 or XFM orthotopic tumors, as the replication of the virus is highly attenuated in murine cell lines. As expected, treatment with the virus did not present a survival benefit in this model (Log-rank test; *P* = 0.98 and *P* = 0.52, for NP53 and XFM, respectively) (Fig. 5k and Supplementary Fig. 6h). These data underscore the importance of the immune response in the antitumor effect of the virus.

In summary, Delta-24-RGD infects, replicates, and kills DIPG murine cells in vitro. Of importance, the in vivo administration

triggers an immune response against DIPG tumors, leading to a significant increase in survival.

## Discussion

The potential of oncolytic adenoviruses in the treatment of brain tumors has been shown by various studies. In fact, several clinical trials have evaluated the suitability of different oncolytic viruses in adult patients with glioblastoma and showed promising results[14,20,21]. The ability of oncolytic virotherapy to trigger an immune response and the possibility of developing new armed viruses with molecules that modulate the immune response have opened new opportunities in cancer therapy

Regarding pediatric brain tumors, several preclinical studies published during the prior 5 years have evaluated the efficacy of different oncolytic viruses[22–24], and most of these works focused on neuroblastoma[25]. For example, the intratumoral injection of HSV1716, which is an oncolytic herpes virus, showed safety and evidence of an immune response and viral replication in young cancer patients[26]. In all of these studies, the oncolytic virus administration displayed no toxicity at the specific doses administered to the pediatric patients. Unfortunately, studies performed in pHGG and DIPG are limited[27], and the use of oncolytic adenoviruses in high-grade pediatric brain tumors continues to be vastly unexplored.

Delta-24-RGD has been widely studied in preclinical and, recently, clinical trials (DNX-2401), demonstrating its safe profile and efficacy against gliomas, but almost all studies were performed in adult patients (NCT00805376, NCT01956734, NCT02197169, NCT02798406). Recent studies have shown that pHGG, including DIPG, differ from their counterparts in adults[3]. Therefore, assessing Delta-24-RGD efficacy in these tumors is timely, as therapeutic options remain suboptimal. In this work, we evaluate the suitability of Delta-24-RGD for the treatment of pediatric gliomas, specifically pHGG and DIPG. Our data show that all cell lines are susceptible to treatment with Delta-24-RGD. We observed that the adenovirus had better antitumor effects in pHGG and was correlated with a tendency to have a higher replication in these cell lines. In Supplementary Fig. S2c there was a statistically significant difference in the replication between the DIPG cell lines and the pHGG cells displayed. Delta-24-RGD contains a 24 bp deletion in the E1A region; thus, viral replication preferentially occurs in cells with a mutation in the RB pathway. Genetic studies have shown that the RB1 mutation is more frequent in hemispheric high-grade gliomas than brain stem tumors, which might explain the higher susceptibility to Delta-24-RGD replication and the higher cytotoxic effect observed in the pHGG cells compared with those in the DIPG cells. In fact, Delta-24-RGD has the strongest antitumor effect against PBT-24 cells, which is a pediatric cell line derived from a child with hemispheric high-grade glioma in which the RB protein was completely absent. Regarding DIPGs, in an elegant study performed by Becher's group they showed that H3.3K27M repressed p16 tumor suppressor (CDKN2A), leading to an increase in proliferation[28]. In addition, other study has shown that PDGF, highly expressed in DIPGs, has the ability to stabilize E2F1[29]. Therefore, even though maybe the RB pathway is not always aberrant at the genomic level (deletions of RB or p16/CDKN2A, amplifications of cyclins or CDKs), there is always E2F1 free owing to uncontrolled replication of tumor cells. Therefore, this fact warrants viral replication[30].

Recent studies have shown the importance of the adenovirus triggering of the immune response in the antitumor response achieved in patients. The lack of relevant models of these types of tumors until recently has also hindered the development of more-tailored approaches. The development of immunocompetent

models that mimic the disease, at least partially[31], have allowed us to test our approach. Our study is important as we evaluate the immune response triggered by an oncolytic adenovirus in two different DIPG murine models. Our results show that Delta-24-RGD administration in the pons of animals is safe and that oncolytic adenovirus administration unleashes a potent immune infiltration in the brain that is mainly circumscribed in the tumor mass, most likely leading to tumor recognition by the immune system. Our results are consistent with data obtained by Dr. Juan Fueyo's group[14,32]. The capacity of Delta-24-RGD to increase lymphocyte infiltration in the brain opens the opportunity for developing new combined strategies with immunotherapy. Immunotherapy developed against DIPG has shown limited results in patients owing to its incapability to cross the blood–brain barrier and reach the tumor along with the strong immunosuppressive microenvironment found in brain tumors[33]. Among the immunotherapies that are currently being test in solid pediatric cancer, are the immune checkpoint inhibitors and the CAR T cells[34]. In addition to insufficient therapeutic effects, treatment using antibodies blocking CTLA-4 have induced the same side effects in pediatric patients than in adults together with insufficient therapeutic effects[35]. In the case of treatment utilizing CAR T cells, despite the encouraging responses observed in liquid tumors, the therapeutic effect has been modest in solid tumors and severe side effects have been reported including (1) the cytokine release syndrome also found in therapeutic monoclonal antibody[36,37] (2) neurological toxicity, patients receiving CD19-specific CAR T cells presented symptoms of neurological damage[38], and (3) on-target/off-tumor recognition, when selected targets of CAR T cells are also expressed in normal tissue; a fatal side-effect example was a patient treated with HER-2/neu CAR T cell that developed a respiratory failure causing patient death[39]. In contrast, administration of DNX-2401 in the context of clinical trials did not show any severe adverse event in either adults[14] nor in DIPG patients[40](and our unpublished results). All together, the therapeutic effect of the Delta-24-RGD is a combination of the virus replication capacity and the anti-tumor effect triggered by it.

The encouraging results obtained in this work supported the opening of a phase I/II clinical trial at our institution to evaluate the safety and efficacy of Delta-24-RGD administration followed by RT in patients with newly diagnosed DIPG[15]. Importantly, as a proof of principle, the administration of Delta-24-RGD has proven to be safe in the first cases[40] (and our unpublished results), with no further toxicity associated with either the biopsy or injection of the virus, supporting the feasibility of this treatment. The lack of severe side effects in this trial is in accordance with the other Delta-24-RGD clinical trials where no adverse effect grade IV has been recorded[14,41].

In summary, here, we present evidence that Delta-24-RGD could constitute a therapeutic option for the treatment of pHGG and DIPG. The use of oncolytic viruses for the treatment of pediatric brain tumors alone or in combination with other strategies could provide a breakthrough in the treatment of these lethal diseases.

## Methods

**In silico analyses of pHGG and DMG/DIPG patient's samples**. Gene expression data from 220 pHGGs and diffuse midline H3-K27M gliomas were collected and integrated as previously described[1]. In brief, gene expression data from an Affymetrix U133 plus 2 microarray, Agilent 44K transcriptome arrays and RNA sequencing were annotated, normalized, and centered independently before being combined, loss renormalized to remove platform bias and median centered. The gene expression data were annotated by subgroup based on the integration of the whole-genome and whole-exome sequencing and application of the Heidelberg methylation classifier for CNS tumors, resulting in six major subgroups (H3.3G34R, H3.3K27_M, H3.1K27_M, IDH1, BRAF_PXA, and

HGG_GBM_H3wt). To compare the absolute levels of expression in the Affymetrix microarray data, rank measurements of the expression of cell surface markers were compared with the present/absent thresholds of gene expression following RNA and mass normalization in the affy package in R.

**Cell lines and culture conditions**. Pediatric glioma CHLA-03-AA cells were obtained from the American Type Culture Collection (ATCC, Manassas, VA). CHLA-200 and SJ-GBM2 were obtained from the Children's Oncology Group (COG Cell Line & Xenograft Repository). The PBT-24 pHGG cell line was developed from a biopsy obtained at the University Clinic of Navarra from a 13-year-old boy. The tumor samples were obtained with signed informed consent. Ethical approval for the obtaining tumor samples was granted by the Ethical Committee of the University of Navarra (CEI; Comité Etico de Investigación) under the protocol numbers CEI-UN 2016-014. Tumors were cut into small pieces, and the cells were enzymatically disaggregated. The cells obtained from the dis-aggregation were cultured with Rosewell Park Memorial Institute medium supplemented with 10% fetal bovine serum and 1% antibiotic. The cell lines obtained from the ATCC or COG were cultured following the manufacturer's specifications. The SF188 cell line was kindly provided by Dr. Chris Jones (Cancer Research Institute, Sutton, UK). The DIPG cell lines TP54, TP80, TP83, and TP84 were kindly provided by Drs. Marie-Pierre Junier and Hervé Cheneiwess (INSERM Institute, Paris, France); the SU-DIPG IV cell line was a kind gift from Dr. Michelle Monje (Stanford, California); and JHH-DIPG1 and JHH-DIPG16A were generously provided by Dr. Eric Raabe (John Hopkins, Baltimore). All DIPG cell lines were maintained as neurospheres cultured in a specialized serum-free basal medium supplemented with a human neural stem cell proliferation supplement (NeuroCult™ NS-A Proliferation Kit, #05751, STEMCELL Technologies), basic fibroblast growth factor and epidermal growth factor (20 ng/mL Sigma-Aldrich, St Louis, MO). All cells were maintained in a humidified atmosphere containing 5% $CO_2$ at 37 °C. All cell lines were routinely tested for mycoplasma (Mycoalert mycoplasma detection kit; Lonza) and authenticated at the CIMA Genomic Core Facility (Pamplona, Spain) using DNA profiling. A description of the cell lines, including the H3 mutational status, has been summarized in the Supplementary Table 1.

The murine DIPG cell lines NP53 and XFM were provided by Dr. Becher (Northwestern University, Illinois, USA)[17,18]. The cell lines were generated from DIPG arising in genetically modified mice. The NP53 cell line was generated from a tumor arising in a DIPG mouse model induced by PDGF-B signaling, TP53 loss, and ectopic H3.3K27M[17]. The XFM cell line was generated from a tumor arising in a mouse model driven by PDGF-B signaling and INK4A and ARF loss[18].

**Viral replication assays**. The pHGG and DIPG cells were seeded at a density of $2 \times 10^5$ cells per well in six-well plates and infected with 10 MOIs of Delta-24-RGD. After 3 days, the cells were collected, and the final amount of virus was determined by a method based on anti-hexon staining in HEK293 cells[42].

**Cell viability assay**. The TP54, TP80, TP83, TP84, SU-DIPG IV, JHH-DIPG-1, and JHH-DIPG-16A cells were seeded at a density of $1 \times 10^4$ cells/well, and PBT-24, CHLA-03-AA, CHLA-200, and SJ-GBM2 were seeded at a density of $2 \times 10^3$ cells per well in 96-well plates. On the following day, the cells were infected with Delta-24-RGD at different MOIs (5, 10, 25, 50, 100, and 300). The cell viability was assessed after 5 days using an MTS assay (Promega) as previously described[43]. The dose–response curves were analyzed using GradPath software. The $IC_{50}$ value is the median-effect dose (dose affecting 50% of the cells, i.e., 50% survival).

**Immunoblotting**. For the immunoblotting assays, the samples were subjected to sodium dodecyl sulphate-Tris-glycine gel electrophoresis. The membranes were incubated with the following antibodies: E1A, (1:1000; sc-430, Santa Cruz Bio-technology, Santa Cruz, CA), fiber (1:1000; (4D2), NB600-541 Novus Biological, Denver, CO), and GRB-2 (1:1000; 610111, BD biosciences). The membranes were developed according to Amersham's enhanced chemiluminescence protocol.

**Animal studies**. Ethical approval for the animal studies was granted by the Animal Ethical Committee of the University of Navarra (CEEA; Comité Etico de Experimentación Animal) under the protocol numbers CEEA/069-13 and 094-15. All animal studies were performed at the veterinary facilities of the Center for Applied Medical Research in accordance with institutional, regional, and national laws and ethical guidelines for experimental animal care. For the orthotopic supratentorial model, CHLA-03-AA and PBT-24 pHGG cells ($5 \times 10^5$) were engrafted by injection into the caudate nucleus of athymic mice. The TP80 and TP54 cells ($5 \times 10^5$) developed DIPG tumors by injecting these cells into the pons of athymic mice, and in both models, a guide-screw system was used[44,45] (Taconic Farms, Inc). The NP53 cells ($10^4$) were implanted in transgenic mice kindly provided by Dr. Oren Becher, and the XFM cells ($10^4$) were implanted in BALB/c female mice. The cells were administered in 3-4 μl of PBS. The animals were randomly assigned to the following two groups: control mice injected with PBS and mice injected with Delta-24-RGD. Delta-24-RGD ($10^8$ pfu per animal) was administered intracranially once or three times (as indicated) in 3–4 μl 3 days after the cell implantation. In general, we consider long-term survivors those animals that live at least two times longer

than the median survival of the control animals. In the case of murine models, in which kinetics are very fast, we consider long-term survivors those animals that live at least three times longer than the median survival of the control animals.

**Immunohistochemical analysis**. Paraffin-embedded sections of mice brains were immunostained for antibodies specific to adenoviral mouse-hexon (1:1000; AB1056, Millipore), adenovirus rabbit-E1A, (1:1000; Santa Cruz Bio-technology, Santa Cruz, CA), CD3 (1:300; clon SP7, NeoMarkers, Fremont, CA), CD4 (1:1000; EPR19514, ab183685 Abcam, Cambridge, MA) CD8a (1:1000, (D4W2Z) #98941 Cell Signaling, Danvers, MA) and FoxP3 (1:400; clon JFK-16s, ref. 14–5773 eBiosciences, Thermo Fisher, Waltham, MA) following conventional procedures. For the immunohistochemical staining, Vectastain ABC kits (Vector Laboratories Inc., Burlingame, CA) were used according to the manufacturer's instructions.

**Statistical analysis**. For the in vitro experiments, the data are expressed as the mean ± SD, and the comparisons were performed using two-tailed Student $t$ tests. The in vivo cytopathic effect of Delta-24-RGD on pHGG and DIPG xenografts was assessed by plotting survival curves according to the Kaplan–Meier method. The survival in different treatment groups was compared using a log-rank test. The program GraphPad Prism 5 (Statistical Software for Sciences) was used for the statistical analyses.

## Data availability

The data that support the findings of this study are available within the paper or Supplementary Information, or source data file or available from the corresponding author upon request.

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

## Acknowledgements

We are very grateful to Dr. Laura Guembe and the imaging core for the help with all the immunohistochemistry. This work was supported by the European Union (Marie Curie IRG270459; to M.M. Alonso), the Instituto de Salud Carlos III y los Fondos Feder Europeos (PI13/125; PI16/0066 to M.M. Alonso), the Spanish Ministry of Science and Innovation (Ramón y Cajal contract RYC-2009–05571, IEDI-2015-00638, and BIO2015-68990-REDT to M.M. Alonso), the Department of Health of the Government of Navarra (to M.M. Alonso), the Basque Foundation for Health Research (BIOEF, BIO13/CI/005), Foundation LA CAIXA/Caja Navarra (A-PG, MMA), Foundation "El sueño de Vicky", Asociation Pablo Ugarte-Fuerza Julen (A-PG,MMA), and DOD team science award (MMA, JF, and CG-M). The Cancer Prevention and Research Institute of Texas (RP170066; C-GM and JF, the Rory David Deutsch Foundation (OJB) and Instituto de Salud Carlos III—CIBERONC (to JAM-C and MJG-B). This project has received funding from the European Research Council (ERC) under the European Union´s Horizon 2020 research and innovation programme (grant agreement No. 817884; ViroPedTher).

## Author contributions

Conception and design: S.T.-S., R.D.-V., J.F., C.G.-M., A.P.-G. and M.M.A. Development of methodology: N.M.-V., M.G.-M., M. Marigil, G.A., S.T.-S., R.D.-V. and M.M.A. Acquisition of data (provided animals, acquired and managed patients, provided facilities, etc.): All authors. Analysis and interpretation of the data (e.g., statistical analysis, biostatistics, and computational analysis): N.M.-V., C.J., A.M. and M.M.A. Writing, review, and/or revision of the manuscript: All authors. Administrative, technical, or material support (i.e., reporting or organizing data and constructing databases): N.M.-V., M.G.-M., M.G.-H., M.P., J.G.P.-L., M.Z., L.M., M.V.-G., V.L., J.J.A., L.I.R., J.A.M.-C., M.J. G.-B., G.A., H.J., M.P.J., H.C., E.A.E.-H., M.M.-D., O.J.B., E.R., A.P.-G. and M.M.A. Study supervision: A.P.-G., C.G.-M., J.F. and M.M.A.

## Additional information

**Competing interests:** H.J., C.G.-M. and J.F. report ownership interest (including patents) in and are consultants for DNATrix. M.M.A., S.T.-S. and R.D.-V. report DNATrix sponsored research not related with this article. The remaining authors do not have potential conflicts of interest to disclose.

