## [Peer Review File · Nature Communications]

Reviewers' comments:

Reviewer #1 (Remarks to the Author):

In this study, the authors have used the human pediatric glioma cell lines CHLA-200, SJ-GBM2, PBT-24, SF-188, and DIPG cell lines TP54, TP80, TP83, TP84, SU-DIPG IV, and JH DIPG 16A, as well as the murine cell lines NP53 and XFM, to test the efficacy of the oncolytic virus Delta-24-RGD, at inducing immunogenic cell death and enhancing cell viability. Orthotopic murine models were generated and used to show the safety of oncolytic virotherapy as well as increased OS. Generated data have already led to an ongoing clinical trial, emphasizing the importance of this research and its clinical-translational significance.

I have major and minor comments with regards to this manuscript:

Major comments:

Methods: Given the importance of the cell lines and mouse models to this work, a better definition of the cell lines and mouse models should be provided. For example the histone mutation status of primary cells is not disclosed. Furthermore, tumor engraftment in murine models should be briefly discussed.

Results line 255: What does "all these deletions" refer to? If it refers to CDKN2A, then given that it is not highly lost in H3.3 and H3.1 DIPGs is contrary to the author's findings. Given that DIPG H3.3 and H3.1 constitute over 70% of DIPG cases, the low deletion rate of CDKN2A deserves further explanation in the context of their findings.

Results line 255: The connection between E2F1 and their finding is not clear. Please expand and explain.

Results line 298: What does "on day 3 of experiment" refer to? How many days post-injection of cells? 3 days? Was this the case for all of the murine experiments?

Results line 303: Define long-term survivors.

Results line 350: I have to disagree with the observation that "the animals displayed a constant weight loss". In fact the weight loss (Figure 5A) does not seem to be constant at all. Please explain.

Results line 366:

Discussion: Discussion should be expanded to explain why treated mice eventually die. Describe this in the context of the ongoing trial and the potential side effects (and potentially deadly) of immunotherapy in general and in their study in particular.

Discussion line 399-405: Virus could be effective against RB mutant HGGs, but not against brainstem gliomas such as DIPGs, if they lack RB pathway mutations. Discussion should not claim that this is a promising therapy for patients with newly diagnosed DIPG, without specifying that its efficacy is dependent on RB mutation status.

Minor comments:

Line 109-11: please rephrase to clearly state the intended point.

Line 134: what is meant by "treatment of Naïve DIPGs"? "Treatment naïve" DIPGs maybe?

Line 147: please correct "H3.3K27_M".

Line 147: what is HGG-GBM-wt? Wild type for what?

Line 175: reference is missing

Line 398: change 24pb to bp

Reviewer #2 (Remarks to the Author):

This manuscript describes the application of Delta-24-RGD, in clinical trials for adult glioma, to pediatric HGG and DIPG, which lack any effective treatments. They demonstrate the susceptibility of a large cohort of human pHGG/DIPG cells and tumors to Delta-24-RGD, validating translation. A

concern with the in vivo studies, that employ 4 representative orthotopic models, is the very early time of treatment, likely before tumors are established. They extend these studies to 2 recently described mouse DIPGs from transgenic mice. Unfortunately, the immunocompetent studies do not include any efficacy outcomes.

Specific Comments,

1. It would make sense to include the data in Fig S2 and S3 in the graphs in Fig 2.
2. The subgroup classification of the cell lines (as in Fig S2A) should be indicated in Fig 2.
3. In Fig 2C, there seems to be a large difference in the levels of fiber protein synthesis between different cell lines even though the infectivity seems quite similar, is there an explanation for this? There is only a limited dose response for E1a expression, is this expected?
4. In Fig 2D and S2C, are there any statistically significant differences between the groups?
5. In Fig S2C the dashed line of input dose is missing. There are 2 bars for TP84, and none for TP80, is one a typo?
6. Why were tumors treated so early after implantation (day 3; Fig 3), especially when most of the tumors are very slow growing, so not treating established tumors? Would this impressive efficacy occur after treatment of established tumors, as would be the case in patients?
7. Indicate the number of mice/group in Fig 3A. The treated mice for TP80 seem to fall into 2 groups, no effect and long-term survivors. Is it possible the no effect group didn't get virus in the tumor, are all the tumors Ad IHC positive (Fig 3H)?
8. Indicate in Fig 3F, G legends when these brains were harvested. Are the sections in Fig 3H from the brains in Fig 3F, G?
9. Don't see any vimentin staining in Fig 3F, only H&E.
10. In Fig 3H, it would be useful to include a low power image of the Ad IHC to illustrate the distribution of Ad-infected cells in the tumor.
11. For Fig 4C, are there any significant changes in titer compared to C? What was the input virus dose (in Fig 2D, MOI 10 was equivalent to 10⁶ pfu/ml)? The IC₅₀ for XFM is similar to that for TP80, yet virus replication is much less, are the mouse cells more susceptible to Ad killing? The demonstration of virus replication would be strengthened by also looking at DNA copy number.
12. What MOI were cells infected with in Fig 4D? Are there EM images for infected XFM cells?
13. For the toxicity studies (Table S1, S2): Was there any histopathology in the brains? What does toxicity mean; death, weight loss, or something else? What are the growth kinetics of NP53? Was the experiment stopped at day 15, a relatively short time, because of tumor growth? Mice are not the best animal to test adenovirus toxicity. For toxicity studies, it would be useful to include wild-type Ad5 as a positive control.
14. In Fig 5B, S4A, it would be informative to see IHC of Hexon and E1a (as in Fig 3H).
15. Are any of the groups significantly different in Fig 5G? It would be useful to include the mean fold-change, as it is not clear that CD4 is greater than PBS (also for Fig S4D).
16. What are the p-values in Fig 5H, S4E?

17. On Ln 361 should be Supp Fig 4D, not 3C.

18. While the increased infiltration of T cells is impressive, in the absence of any data on survival/tumor growth in the immunocompetent models, the relevance of these TIL changes to therapy is unclear.

Reviewers' comments:

Reviewer #1 (Remarks to the Author):

In this study, the authors have used the human pediatric glioma cell lines CHLA-200, SJ-GBM2, PBT-24, SF-188, and DIPG cell lines TP54, TP80, TP83, TP84, SU-

DIPG IV, and JH DIPG 16A, as well as the murine cell lines NP53 and XFM, to test the efficacy of the oncolytic virus Delta-24-RGD, at inducing immunogenic cell death and enhancing cell viability. Orthotopic murine models were generated and used to show the safety of oncolytic virotherapy as well as increased OS. Generated data have already led to an ongoing clinical trial, emphasizing the importance of this research and its clinical-translational significance. We thank the reviewer for these kind words highlighting the importance of our study in pHGG and DIPG.

I have major and minor comments with regards to this manuscript:

Major comments:

Methods: Given the importance of the cell lines and mouse models to this work, a better definition of the cell lines and mouse models should be provided. Most of the cell lines have been developed by other groups and their characterization has been published by them. Nevertheless, we have included a table (Supplementary Table 1) where we included the sex, the age, the H3 mutational status and a reference for each of the cell lines used in this study.

For example the histone mutation status of primary cells is not disclosed. -As mentioned above, we have added a table with the description of the H3 mutation status and this information is included as Supplementary Table 3.

Furthermore, tumor engraftment in murine models should be briefly discussed. - Before we perform survival experiments we assess the engraftment and the kinetics of the different cell lines in their different mouse strains. Regarding the engraftment technique for supratentorial tumors we have followed the protocol published by Lal et al; Neurosurg 2000 and for the DIPG engraftment we have followed the protocol developed by our lab; Marigil et al., Plos One 2017. Bellow, we have detailed the full references for your information. These references are also included in the article. Nevertheless, we have added a more detailed description of the techniques in the supplemental text.

-Lal S, Lacroix M, Tofilon P, Fuller GN, Sawaya R, Lang FF. An implantable guide-screw system for brain tumor studies in small animals. J Neurosurg. 2000;92:326–33.

-Marigil M, Martinez-Velez N, Domínguez PD, Idoate MA, Xipell E, Patiño-García A, et al. Development of a DIPG orthotopic model in mice using an implantable guide-screw system. PLoS One. 2017;12

Results line 255: What does “all these deletions” refer to? Yes, the reviewer is right and refers to the different tumor types as described in the article by McKay et al Cancer Cell. 2017;32:520–537.e5. ***If it refers to CDKN2A, then given that it is not highly lost in H3.3 and H3.1 DIPGs is contrary to the author’s findings. Given that DIPG H3.3 and H3.1 constitute over 70% of DIPG cases, the low deletion rate of CDKN2A deserves further explanation in the context of their findings.*** -This is an interesting point raised by the reviewer and it is true that RB pathway genomic aberrations are not highly abundant However, E2F1, a transcription factor that is downstream of this pathway and drives the expression of the adenoviral program in

highly expresses. The natural host of the wild type adenovirus are the epithelial upper respiratory tract cells, which are quiescent. The virus in order to replicate needs cycling cells. To do so, the virus expresses the E1A protein (a master regulator of the viral program) that in turn binds RB releasing its brake from E2F1 and allowing these cells to enter the cell cycle. Viral replication is dependent in the presence of E2F1, a transcription factor, which moves the cells to S phase and therefore moves all the cell machinery to work. This transcription factor is abundant in all cancer cells since they are cycling. Therefore, even though maybe the RB pathway is not always aberrant at the genomic level (deletions of RB or p16/CDKN2A, amplifications of cyclins or CDKs) there is always E2F1 free due to uncontrolled replication of tumor cells. This fact, permits viral replication. We have clarified this in the discussion section. For further information there is a very good review from Gomez-Manzano et al., 2004 and this reference has been included in the discussion (lines 434-439).

Figure 3. Illustration of the proposed mechanism for the selective replication of the E1A-mutant Delta-24 adenovirus in cancer cells.

Candelaria Gomez-Manzano et al. Neurology 2004;63:418-426

Figure 3. Illustration of the proposed mechanism for the selective replication of the E1A-mutant Delta-24 adenovirus in cancer cells. (A) In the majority of normal cells in the adult brain, retinoblastoma (Rb) binds and represses transcription factors, such as E2F, arresting the cells in G₀. (B) When wild-type adenoviruses infect normal cells, E1A adenoviral protein binds to and inactivates Rb. This interaction results in the release of E2F, and the cell cycle progresses with unscheduled DNA replication creating the optimal milieu for viral replication. (C) However, when the E1A-mutant adenovirus infects normal cells, the wild-type Rb protein remains unaffected. As a result, the normal cells remain arrested in G₀, halting the replication of the mutant adenovirus. (D) If an E1A-mutant adenovirus infects cancer cells with a disrupted Rb pathway, the adenovirus will be modulated by free E2F activity. These favorable conditions in the cells foster cell cycle progression and subsequent efficient replication that results in cell death and production of new adenoviral progeny.

Results line 255: The connection between E2F1 and their finding is not clear. Please expand and explain. -See above

Results line 298: What does "on day 3 of experiment" refer to? It means 3 days post cell injection. **How many days post-injection of cells? 3 days?** Yes, that is

correct. **Was this the case for all of the murine experiments?** Yes, this the case for all the murine experiments, except for the new experiments with the established tumors, which have been treated on day 15 post cell injection. The information regarding the treatment day has been included in the figure legends for clarification.

Results line 303: Define long-term survivors. In general, we consider long-term survivors those animals that live at least 2 times longer than the median survival of the control animals. In the case of murine models which kinetics are very fast we consider long-term survivors those animals that live at least 3 times longer than the median survival of the control animals.

Results line 350: I have to disagree with the observation that “the animals displayed a constant weight loss”. In fact the weight loss (Figure 5A) does not seem to be constant at all. The reviewer is right and we have changed this statement. **Please explain. Results line 366.** –Basically, splenocytes isolated from Delta-24-RGD treated mice produced significantly higher amounts of INFgamma than those from PBS-treated mice. These results strongly suggest the existence of a specific antitumor immune response. We have rephrase the sentence to clarify.

Discussion: Discussion should be expanded to explain why treated mice eventually die. Treated mice eventually died because not all of them respond in the same way to the treatment, albeit all the conditions are the same. This fact might be due to several tumor and/or mice intrinsic reasons (For example, necrotic or fibrotic tumors could be a barrier to virus replication) and extrinsic reasons (For example, it could be that due to human error some animals might receive less dose of virus or less tumor cells were engrafted). Since, our discussion is already fairly long we have not included this information in it. We apologize for the inconvenience. **Describe this in the context of the ongoing trial and the potential side effects (and potentially deadly) of immunotherapy in general and in their study in particular.** Immunotherapy is a very broad term encompassing many different approaches, from vaccines to immune check-point or CAR-T cells all of them with different type of side-effect. Virotherapy has unique side effects that also can vary depending on the type of virus used. In that sense, is not the same a poliovirus than an adenovirus. In my experience with Delta-24-RGD (an adenovirus) in our clinical trials the side effects of the virus has been extremely mild. With the most severe effect observed being fever. In our trial there has not been any severe adverse effect. Even in the clinical trial with the DIPGs (so far 7 patients enrolled and 5 treated off label) we have not observed any fatal nor severe adverse effect. We have included a paragraph in the discussion to comment on this (lines 463-465).

Discussion line 399-405: Virus could be effective against RB mutant HGGs, but not against brainstem gliomas such as DIPGs, if they lack RB pathway mutations. Discussion should not claim that this is a promising therapy for patients with newly diagnosed DIPG, without specifying that its efficacy is dependent on RB mutation status. –We agree with the reviewer that this point has not been well explained and we have rephrased it in the discussion. The virus, as mentioned above, needs free E2F in order to activate some pivotal viral promoters. We can say that free E2F1 is nearly a universal feature this signaling pathway is always deregulated because one of the hallmarks of cancer cells is constant proliferation.

Therefore, virus will nearly always replicate. Another trait of the virus is that its therapeutic effect not only relies in the proliferation of the virus but also in the antitumor immune response that it is able to trigger. Therefore, in slow cycling cells even though the virus will take longer to act eventually it will replicate and it will trigger the immune response. We have added a paragraph to the discussion (lines 434-439 and 457-459).

Minor comments:

Line 109-11: *please rephrase to clearly state the intended point.* We have rephrase the sentence to clarify our point.

Line 134: *what is meant by “treatment of Naïve DIPGs”? “Treatment naïve” DIPGs maybe?* We have changed to the sentence to newly diagnosed DIPGs to avoid confusion.

Line 147: *please correct “H3.3K27_M.* We have corrected the nomenclature.

Line 147: *what is HGG-GBM-wt? Wild type for what?* We meant wild type for H3. We have also corrected it in the text to clarify it.

Line 175: *reference is missing* . The reference was included further down with the more detailed discussion of the cell lines. However, as requested by the reviewer we also included the references there. The reviewer is right and helps to understand from where comes each of the cell lines.

Line 398: *change 24pb to bp.* We apologize for the typo and we have corrected it in the text and highlighted in red. Due to the changes now is in line 409.

Reviewer #2 (Remarks to the Author):

This manuscript describes the application of Delta-24-RGD, in clinical trials for adult glioma, to pediatric HGG and DIPG, which lack any effective treatments. They demonstrate the susceptibility of a large cohort of human pHGG/DIPG cells and tumors to Delta-24-RGD, validating translation. A concern with the in vivo studies, that employ 4 representative orthotopic models, is the very early time of treatment, likely before tumors are established. They extend these studies to 2 recently described mouse DIPGs from transgenic mice. Unfortunately, the immunocompetent studies do not include any efficacy outcomes. -We thanks the review for her/his insights and we agree with the notion that the study would benefit from the evaluation of the therapeutic efficacy of the virus in established tumors and from efficacy data in immunocompetent mice. In order to address his/her concerns we have performed in vivo studies in immunodepressed mice after the establishment of tumors. This new set of data has been included in Figure 3. In addition, we have data have been added in Figure 5.

Specific Comments,

1. It would make sense to include the data in Fig S2 and S3 in the graphs in Fig 2. -We thank the reviewer for his/her suggestion and we understand his thinking. However, due to space constraints we believe it is easier to read the figure as it is.

2. The subgroup classification of the cell lines (as in Fig S2A) should be indicated in Fig 2. -We have included this information in Fig.2

3. In Fig 2C, there seems to be a large difference in the levels of fiber protein synthesis between different cell lines even though the infectivity seems quite similar, is there an explanation for this? -Because once inside of the cells the virus produces such a big amount of viral proteins in many instance the blot gets saturated and therefore is difficult to find a dose dependent effect. Western blot is not sufficiently sensitive to detect subtle difference amongst virus treated cells.

There is only a limited dose response for E1a expression, is this expected? -As we explained above, because the extremely big amounts of viral proteins produced inside of the cells in many instance they saturated the membrane and is hard to see a dose dependent effect.

4. In Fig 2D and S2C, are there any statistically significant differences between the groups? -In figure 2D there is a statistically significant difference between the DIPG cell lines and the pHGG cells displayed ($P < 0.01$). However, there is no difference amongst the cell lines displayed in Supplementary Figure S2C. The replication varies amongst the different cell lines and we do not see a clear pattern between DIPG and pHGG cell lines. Although, in general, the titer of the virus is slightly higher in pHGG cell lines than DIPGs.

5. In Fig S2C the dashed line of input dose is missing. -The reviewer is right; we have added the dashed line showing the input (10^6 pfu). **There are 2 bars for TP84, and none for TP80, is one a typo?** -Yes, it is a typo. We apologize for the mistake and we have included the right name for the cell line, which is the TP80 as the reviewer suggested.

6. Why were tumors treated so early after implantation (day 3; Fig 3), especially when most of the tumors are very slow growing, so not treating established tumors? Would this impressive efficacy occur after treatment of established tumors, as would be the case in patients? -This is a very good point raised by the reviewer and generally speaking because the aggressiveness of the brain tumor we start to treat 3 days after injection. In fact, by that day there is already visible tumor as assess by IHC in many of the cell lines used. As an example, we have added representative images of the cell line CHLA-03-AA at 3, 7 and 15 days, PBT-24 at 15 days and Tp54 at 15 days. These results have been included in Supplementary Figure 4H.

Nevertheless, we agree with the reviewer that it would be interesting to assess the therapeutic effect in established tumors and we have performed new experiments with the CHLA-03-AA and PBT24 treating the tumors 15 days later. These data have been included in Figure 3H. As it can be observed in the below survival curves treatment of established tumors (15 days post-injection) with Delta-24-RGD results in a significant increase in survival in both cell lines ($P=0.005$ and $P=0.04$ for CHLA-03-AA and PBT-24, respectively). As expected, the result although good is not as impressive as the survival curves observed in tumors where we start to treat 3 days post-injection.

7. Indicate the number of mice/group in Fig 3A. -There were 8 animals in the control group and 9 in the Delta-24-RGD-treated group.

The treated mice for TP80 seem to fall into 2 groups, no effect and long-term survivors. Is it possible the no effect group didn't get virus in the tumor, are all the tumors Ad IHC positive (Fig 3H)? -Although, it might be a possibility a human error our experience is that not all mice respond in the same way to the treatment, albeit all the conditions are the same. These is due to several tumor/mice intrinsic reasons (For example, necrotic or fibrotic tumors could be a barrier to virus replication) and extrinsic reasons(For example, it could be that due to human error some animals might receive less dose of virus or less tumor cells were engrafted). We can safely affirm that in the virus-treated that did not respond or fare worst we still are able to detect hexon and/or E1A. The only mice that do not present immunoreactivity to viral proteins are the long-term survivors with **no tumors**. In the absence of tumor cells, the virus cannot replicate and ceases its activity. Long-term survivor mice with small residual tumors present viral proteins to up to 3 months and longer in some instance. Always viral proteins are circumscribed to the tumor.

8. Indicate in Fig 3F, G legends when these brains were harvested -The information has been included in the figure legends. Brain harvested for comparison come from mice that died at a similar time point from control (PBS) and virus-treated groups: CHLA-03-AA= 50±10 days; PBT-24= 60±5 days; PBT80= 200±5 days; Tp54= 90±5 days. **Are the sections in Fig 3H from the brains in Fig 3F, G?** -Yes, these brains come from mice treated within the experiment, that died at similar times.

9. Don't see any vimentin staining in Fig 3F, only H&E. -Yes the reviewer is right this is an error from our part and we have removed this statement from the paper.

10. In Fig 3H, it would be useful to include a low power image of the Ad IHC to illustrate the distribution of Ad-infected cells in the tumor. -This is a good suggestion and we have included a low power image of the ad IHC to illustrate the distribution of the ad-infected cells. This new figure has been included in Supplementary Figure 4B (DIPG cell lines) and 4C (pHGG cell lines).

B

C

11. For Fig 4C, are there any significant changes in titer compared to C? –The virus replication is very attenuated in mice cells. In fact, generally speaking, adenovirus are specie-specific. Therefore, we need to use high Delta-24-RGD MOI to see a slight replication. Nevertheless, we do observed a significant ($P < 0.05$) increase in replication when we compare the 300 MOI infected cells (NP53 and XFM) in comparison with their control (input virus).

What was the input virus dose (in Fig 2D, MOI 10 was equivalent to 106 pfu/ml)?
Yes, that is correct the total virus quantified in that experiment is 10^6 pfu/ml.

The IC50 for XFM is similar to that for TP80, yet virus replication is much less, are the mouse cells more susceptible to Ad killing? -Yes, the reviewer is right and the XFM cells are much more susceptible to the virus than the NP53. One explanation could be that since the virus readily infects the XFM cells, even though replication is attenuated, big amounts of E1A proteins are produce. E1a has the ability to induce cell death. We also observed this effect in the in vivo experiments where mice bearing XFM cell line and treated with the virus have a significant better overall survival than NP53 bearing mice and treated with the virus.

The demonstration of virus replication would be strengthened by also looking at DNA copy number. – We agree with the reviewer that quantifying virus’s copy number would also showed proof of viral replication. However, we believe that using a functional assay such as the Hexon assay and complemented with the electron microscopy also provides supportive evidence of viral replication. We included representative electron microscopy images of assembled virions in XFM. In the picture, we can also observed huge amounts of viral proteins that are not inserted in functional viruses. We hope that the reviewer finds this information acceptable.

12. What MOI were cells infected with in Fig 4D? 300 MOIs Are there EM images for infected XFM cells? -As we mentioned above, we do have images for XFM and we have included them in Figure 4D.

13. For the toxicity studies (Table S1, S2): Was there any histopathology in the brains? –Yes, we routinely performed IHC of all the brains at least using H&E and we did not find any abnormality or anything that was subjected to be shown.

-What does toxicity mean; death, weight loss, or something else? -We defined toxicity as all of the above death, weight loss, symptoms of suffering or disease. In the past, we have assessed the toxicity after intravenous infection to other organs such as liver, because liver is one of the prefer organs for the virus, spleen, lungs. However, at the dosage used we do not observed toxicity. Moreover, since we performed intratumoral injection we have not seen traces of the virus nor viral replication in these organs. Other parameters that we have also checked in the past are transaminases. Again these enzymes has shown to be within normal parameters.

- What are the growth kinetics of NP53? -Before performing survival studies we always conduct an study with different number of cells to assess the kinetics and find a number of cells that give us also a sufficient time to assess the treatment. We have included the kinetics for the Np53 and XFM in Supplementary Figure6A. We opted to use 10^4 cells per animal due to the bigger window of treatment.

Was the experiment stopped at day 15, a relatively short time, because of tumor growth? -In experiments where we aim to assess the immune response we sacrifice the mice at 15 days to ensure that it is the response is already mounted. When we performed survival studies, the experiment goes longer. Similarly, as we do with the human cell lines, in the murine cell lines we wait at least twice the median survival of the control mice to define long-term responders in the treated group.

Mice are not the best animal to test adenovirus toxicity. For toxicity studies, it would be useful to include wild-type Ad5 as a positive control. Yes the reviewer is completely right and as we mentioned above our group and other has extensively tested the toxicity of wild-type adenovirus (for example see Cascallo et al., Molecular Therapy 2007). Since Delta-24-RGD is already in the clinic, our group and external agencies such as the FDA also check extensively the toxicity of the virus in cotton rats and so far always Delta-24-RGD has proven to be safe.

14. In Fig 5B, S4A, it would be informative to see IHC of Hexon and E1a (as in Fig 3H). We have included images of E1A in Supplementary Figure 6D.

15. Are any of the groups significantly different in Fig 5G? . The data is express as the 2^{-ddCT} being the Control RNA from PBS-Treated mice given a value of 1. Actually the three groups are significantly different from their controls in NP53 and XFM cell lines (Figure 5G and Supplementary Figure **It would be useful to include the mean fold-change, as it is not clear that CD4 is greater than PBS** – The fold change has

been include (also for Fig S4D). **16. What are the p-values in Fig 5H, S4E?** Yes, the difference is significant and the p values have been included in the figures ($P=0.0001$ and $P=0.0005$, respectively). Supplementary Figure 4E now has changed to supplementary Figure 5E.

17. On Ln 361 should be Supp Fig 4D, not 3C. -The reviewer is right and we have corrected this typo and renumbered the figures to accommodate all the changes.

18. While the increased infiltration of T cells is impressive, in the absence of any data on survival/tumor growth in the immunocompetent models, the relevance of these TIL changes to therapy is unclear.

-We have performed a new set of survival experiment to assess the antitumor effect of Delta-24-RGD in the immunocompetent DIPGs models using the murine cell lines NP53 and XFM. For the NP53 cell line we conducted 3 different experiments:

1.- First, we injected the NP53 in the flank of syngeneic mice and assess the tumor growth comparing PBS-treated mice vs Delta-24-RGD treated mice. As expected, tumors of Delta-24-RGD grew significantly less than PBS-treated mice. These results have been included in Figure 5I.

2. In a second set of experiment mice bearing orthotopically Np53 or XFM DIPG murine cells were treated with either PBS or Delta-24-RGD ($\text{pfu} = 10^7$). In both models, the virus significantly increase the median overall survival (Being, $P=0.01$ and $P=0.0002$) and led to long term survivors (N=1 for the NP3) and N=8 for the XFM). This new data has been included in Figure5J and Supplemental Figure 6B.

In addition, we subjected the long-term survivors from these two experiments to a rechallenge to assess the possible generation of memory. For the NP53 is still very early to say but for the XFM cell line we already see a trend in which the naïve mice are already dying and not the Delta-24-RGD. We only include these data for the reviewer perusal because due to volume and time-consuming of these experiments they are still on-going and our review time was finishing.

3. Finally, to demonstrate that the efficacy effect in the murine immunocompetent models was mainly due to the immune response generated against the tumor we performed another set of experiments where we orthotopically implanted NP53 and XFM in athymic nude mice. In this model, we did not find any significant difference between the median survival of the PBS-treated mice and the Delta-24-RGD-treated mice ($P=0.98$ and $P=0.52$ for the NP53 and XFM, respectively). These data suggest that in the context of murine cells where the viral replication is very attenuated the triggering of an immune response is very important for the antitumor effect displayed by the virus. This data has been included in Figure 5 K and Supplementary Figure 6C. Both models have been included here to ease the comparison.

Reviewers' comments:

Reviewer #1 (Remarks to the Author):

The manuscript have improved as a result of revisions. However, I have further comments which are mainly concerned the formatting, and the authors response.

1. Refences should be reviewed to ensure that they accurately represent the text.
2. Supplementary Figure 1: The references do not reflect the indicated cell lines. The table should be carefully revised.

3. The author's response to the following question is confusing:

"For example the histone mutation status of primary cells is not disclosed. -As mentioned above, we have added a table with the description of the H3 mutation status and this information is included as Supplementary Table 3. "

Are the referring to Sup Table 1 or 3???

4. In response to Tumor engraftment, the authors responded: "we have added a more detailed description of the techniques in the supplemental text.

I am not sure which supplemental text they are referring to. The answer is not adequate.

5. In response to "Results line 299..." the authors responded: "The information regarding the treatment day has been included in the figure legends for clarification."

Which figure, what line number and how can I easily fine their revisions?

6. In response to: "Results line 350: I have to disagree with the observations", The reviewer is right and we have changed this statement. "

Again, I have a hard time locating the revised text. The same stands for their respond to question on "Results Line 366"

7. In response to: "Describe this in the context of ongoing trials.." the authors responded:

"We have included a paragraph in the discussion to comment on this (lines 463-465)."

Lines 463-465 do NOT discuss immunotherapy at al.

Response to statistical significance of "Fig 2D and S2C," the authors indicate: "In figure 2D there is a statistically significant difference between the DIPG cell lines and the pHGG cells displayed ($P < 0.01$). However, there is no difference amongst the cell lines displayed in Supplementary Figure S2C. The replication varies amongst the different cell lines and we do not see a clear pattern between DIPG and pHGG cell lines. Although, in general, the titer of the virus is slightly higher in pHGG cell lines than DIPGs."

Was this incorporated in the figure/text? If not, it should be.

8. Groups in 5G are significantly different per authors response. This should be reflected in the figure.

9. What is the p-value for the asterisk in 5I?

Reviewer #2 (Remarks to the Author):

The authors have provided important new data and significantly improved the manuscript. The details from many of their responses should be added to the manuscript (see specifics below).

Specifics:

For Reviewer #1:

In 298. Didn't see changes to text (figure legends). Could be clarified on In 298.

In 303. Add definition to text (ie., methods/results)

For Reviewer #2:

3. Not a persuasive argument, the amount of protein loaded or exposure times can be adjusted depending on the band intensity.

4. The statistical differences should be indicated in figure or legend.

7. In Fig S4B, it is difficult to see any hexon+ cells in TP80, is this because there are so few? In Fig S4B, C legends describe experimental details; ie., when was virus injected, when were brains harvested.

8. There are only single brains illustrated for each tumor, so the time of brain harvest cannot be . Indicate the time of brain harvest for each brain in Fig 3E-B legends. Indicate in figure legend that images in Fig 3G are from brains in Fig 3E, F.

11. Indicate statistics in figure 4C legend. In Fig 2D legend indicate that MOI 10 = 10^6 pfu/ml.

13. Indicate in Table S2 how toxicity was measured.

In 331. Should be Fig 3I

In 440. How much expression is relevant amount and what is this based on?

In 699. Should be 15 days post-cell implantation.

Fig 3 A-D, H, I. What does the blue shaded area represent?

February 12, 2019

RE: NCOMMS-D-16-00129_R2

Editor's comment: Please, find the results from the rechallenge experiments performed using NP53 and XFM cell re-implantations in Figure 5J and Supplementary Figure 6G.

Figure 5J

Supplementary Figure 6G

Reviewers' comments:

Reviewer #1 (Remarks to the Author):

The manuscript has improved as a result of revisions. However, I have further comments which are mainly concerned the formatting, and the authors response. We thank the reviewer for his/her comments and we are providing further information to answer the concerns raised.

1. References should be reviewed to ensure that they accurately represent the text. We have reviewed all the references ensuring that they now match the text in revision 2 of the manuscript.

2. Supplementary Figure 1: The references do not reflect the indicated cell lines. The table should be carefully revised. We have revised the references regarding the cell lines.

3. The author's response to the following question is confusing: "For example the histone mutation status of primary cells is not disclosed. -As mentioned above, we have added a table with the description of the H3 mutation status and this information is included as Supplementary Table 3. "Are the referring to Sup Table 1 or 3???" We are sorry about the confusion. We included the details regarding the cell lines in Supplementary Table 1. In the new revision, we have placed the references regarding Supplementary Table 1 in the Supplementary Text file, immediately below Supp Table 1. .

4. In response to Tumor engraftment, the authors responded: "we have added a more detailed description of the techniques in the supplemental text. I am not sure which supplemental text they are referring to. The answer is not adequate. We added the information regarding tumor engraftment in the supplemental text file, in page 7-8. We are also including this information below for the reviewer's perusal.

"Tumor establishment procedure. Under aseptic conditions and with all materials sterilized according to standard techniques, mice of four weeks of age were anesthetized by intraperitoneal injection with ketamine and xylazine solution. The animal heads were supported by a couple of rolled gauzes so that when the screw was inserted, pressure applied over neck and head structures was better tolerated by the animal. We prepared mice head skin with povidone iodine solution prior to make a 5 mm-long lineal skin incision with 23-size scalpel and expose skull sutures. We first made a small mark according to the coordinates with a small 15-gauge needle, which was subsequently widened with a hand-controlled twist drill, which penetrates the skull. Next, we introduced the screw with its specific screwdriver by applying slight pressure throughout the previous twist hole. The coordinates for generation of DIPG tumors are 1.0 mm right to lambda and just posterior (0.8 mm) to lambdoid suture, while coordinates for pHGG tumors are from bregma (intersection between coronal and sagittal suture) 1 mm anterior and 2.5mm to the right. Thereafter the needle of Hamilton syringe is slowly introduced into the hole by applying gentle pressure until the sleeve/cuff from the syringe reaches the screw surface. The desired depth to reach brainstem is 6.5 mm and depth for hemispheric tumors (pHGG) is 3.5mm. Cell suspension was carefully injected using an infusion pump (Harvard Apparatus) over 20 minutes"

5. In response to "Results line 299..." the authors responded: "The information regarding the treatment day has been included in the figure legends for clarification." Which figure, what line number and how can I easily find their revisions? This information is included in the main text of the article page 14 lines 304-305 and highlighted in red: "A single dose (10^8 pfu per animal) of Delta-24-RGD was intratumorally injected on day 3 after cell injection."

In the Figure legend 3, the following statement is included in page 32 line 696: "Animals were treated 3 days post-tumor cell injection."

6. In response to: "Results line 350: I have to disagree with the observations", The reviewer is right and we have changed this statement. "Again, I have a hard time locating the revised text. The following text was added in page 17 lines 371-374 and in this revised version we have highlighted the text in red: "Since weight loss is a sign of toxicity, we monitored this parameter for 15 days after the virus injection. The animals did not display a significant weight variation related with PBS or Delta-24-RGD administration (Supplementary Fig. S5A)".

The same stands for their respond to question on "Results Line 366". In the revised version the following text was added in lines page 17 lines 388-391: "The splenocytes from the Delta-24-RGD-treated mice expressed significantly more IFN γ than those extracted from the PBS-treated mice bearing NP53 or XFM-derived tumors ($P=0.001$, $P=0.0005$, respectively)

(Fig. 5G and Supplementary Fig. S6E), suggesting that there is a specific antitumor immune response”. In addition, for clarity, we highlighted the text in red.

7. In response to: “Describe this in the context of ongoing trials.” the authors responded: “We have included a paragraph in the discussion to comment on this (lines 463-465).” Lines 463-465 do NOT discuss immunotherapy at all. In this revised version we included the following paragraph in the discussion section in page 22 and lines 477-491: “Amongst the immunotherapies that are currently being test in solid pediatric cancer, are the immune checkpoint inhibitors and the CAR-T cells ³⁹. In addition to insufficient therapeutic effects, treatment using antibodies blocking CTLA-4 have induced the same side effects in pediatric patients than in adults together with insufficient therapeutic effects ⁴⁰. In the case of treatment utilizing CAR-T cells, despite the encouraging responses observed in liquid tumors, the therapeutic effect has been modest in solid tumors and severe side effects have been reported including 1) the cytokine release syndrome (CRS) also found in therapeutic monoclonal antibody ^{41,42} 2) neurological toxicity, patients receiving CD19-specific CAR T-cells presented symptoms of neurological damage ⁴³, and 3) on-target/off-tumor recognition, when selected targets of CAR T-cells are also expressed in normal tissue; a fatal side effect example was a patient treated with HER-2/neu CART T cell that developed a respiratory failure causing patient death ⁴⁴. In contrast, administration of DNX-2401 in the context of clinical trials did not show any severe adverse event in either adults ¹⁴ nor in DIPG patients ⁴⁵(and our unpublished results)”

8. Response to statistical significance of “Fig 2D and S2C,” the authors indicate: “In figure 2D there is a statistically significant difference between the DIPG cell lines and the pHGG cells displayed (P<0.01). However, there is no difference amongst the cell lines displayed in Supplementary Figure S2C. The replication varies amongst the different cell lines and we do not see a clear pattern between DIPG and pHGG cell lines. Although, in general, the titer of the virus is slightly higher in pHGG cell lines than DIPGs.” Was this incorporated in the figure/text? If not, it should be. In order to clarify this point, we have placed in Figure 2D the individual results related with viral replication experiments in the different cell lines. In the Supplementary Figure 2C, we performed statistical analysis, by grouping the results obtained in pHGG and DIPG cell lines. This analysis showed that, indeed, there is a small, albeit significant, increase in the replication of Delta-24-RGD in pHGG cell lines when compared with DIPG cell lines (P=0.03). These new data have been incorporated in the results section page 13, line 290-291, and in the discussion section, page 20/21, lines 444-447. (highlighted in red).

9. Groups in 5G are significantly different per authors response. This should be reflected in the figure. Please, note that, in the revised version, Figure 5G is now Figure 5F. The P values have been incorporated to the figure.

10. What is the p-value for the asterisk in 5I? Please, note that, in the revised version, Figure 5I is now Figure 5H. The P value is 0.01 and has been incorporated to the figure.

Reviewer #2 (Remarks to the Author):

The authors have provided important new data and significantly improved the manuscript. The details from many of their responses should be added to the manuscript (see specifics below).

Specifics:

For Reviewer #1:

In 298. Didn't see changes to text (figure legends). Could be clarified on In 298. As indicated above to the reviewer 1, this information is included (highlighted in red) in the main text of the article, page 14, lines 304-305: "A single dose (10^8 pfu per animal) of Delta-24-RGD was intratumorally injected on day 3 after cell injection."

In the figure legends, the following statement is included in page 32, line 696: "Animals were treated 3 days post-tumor cell injection."

In 303. Add definition to text (ie., methods/results). The below definition has been added to the text in the methods section in pg 10, lines 219-223: "In general, we consider long-term survivors those animals that live at least 2 times longer than the median survival of the control animals. In the case of the syngeneic murine models which kinetics are very fast we consider long-term survivors those animals that live at least 3 times longer than the median survival of the control animals."

For Reviewer #2:

3. Not a persuasive argument, the amount of protein loaded or exposure times can be adjusted depending on the band intensity. The reviewer is right, and in general with cellular proteins we have no problem to assess differences. The data presented here showed a dose dependent expression of fiber (TP54, SU-DIPG IV) and E1A (TP54, CHLA-03-AA). However, these cells are very well infected and with 10 MOIs we can detect very high levels of viral proteins even with 5 MOIs. While this is a positive finding, we agree with the reviewer on his/her remarks, and we have modified the text in order accordingly.

4. The statistical differences should be indicated in figure or legend. Please see response to Reviewer #1, section 8.

7. In Fig S4B, it is difficult to see any hexon+ cells in TP80, is this because there are so few? That is correct, there are very few tumoral cells, and therefore the amount of detectable hexon is low. In addition, the persistence of viral replication in long-term survivors is very low. *In*

Fig S4B, C legends describe experimental details; ie., when was virus injected, when were brains harvested. We have included that information in the Supplemental Text page 5 as follows: “Mice were treated with either PBS (control) or Delta-24-RGD 3 days post-cell injection. For comparison studies, analyzed brains are derived from mice that died at a similar time point for both control (PBS) and virus-treated groups: CHLA-03-AA= 50±10 days; PBT-24= 60±5 days; Tp80= 200±5 days; Tp54= 90±5 days.”

8. There are only single brains illustrated for each tumor,.... Indicate the time of brain harvest for each brain in Fig 3E-B legends. The information has been included in the figure legends in Figure 3-E and F page 33, lines 717-720: “Brain from mice that died at a similar time point were analyzed to compare the effect of control (PBS) versus virus-treated groups: CHLA-03-AA= 50±10 days; PBT-24= 60±5 days; Tp80= 200±5 days; Tp54= 90±5 days.” **Indicate in figure legend that images in Fig 3G are from brains in Fig 3E, F.** We have included that information in page 32, line 702-705, and are highlighted in red: “Images correspond to the brains shown in Figure 3E and F.”

11. Indicate statistics in figure 4C legend. For clarity to the reader, we have added the P values to the figure and in the result section, page 16, line 353-355 and we have highlighted it in red: “We observed that the virus replication was significantly higher in NP53 or XFM infected with 300 MOIs (P=0.07 and P=0.005 respectively) when compared with the initial inoculum”

C

12. In Fig 2D legend indicate that MOI 10 = 10⁶ pfu/ml. This information has been added to the Figure 2 legend in page 31, line 681 (highlighted in red).

13. Indicate in Table S2 how toxicity was measured. We have added the following sentence to the Supplementary Table 2: “Toxicity was evaluated by assessing different symptoms including; paresia, gate abnormality, difficulty of access to food or water, weight loss of >20%, or the death of the animal after viral or PBS injection.”

In 331. Should be Fig 3I. We have added this information in line 336 of the revised version.

In 440. How much expression is relevant amount and what is this based on? Because the E2F1 levels presented in the Supplementary Figure 1C are absolute levels I understand that is difficult to justify if they are relevant or not. Normal, healthy astrocytes do not express E2F1 or very little because they are quiescent. In our experience based in our previous work (Alonso et al., JNCI 2005, Alonso et al., Cancer Res 2007) expression levels such as the one displayed by the different molecular entities of pediatric high grade glioma an DIPGs (Supplementary Figure 1C) indicate that cells are proliferating and therefore susceptible to viral replication. In any case, to avoid confusion we have deleted that statement and included the following paragraph in the

discussion where we comment on the possibilities of why in DIPGs even though RB and CDKN2A deletions are not very common there is E2F and therefore viral replication. The statement has been included in the discussion section page 21 lines 455-458: “*Regarding DIPGs, in an elegant study performed by Becher’s group they showed that H3.3K27M repressed p16 tumor suppressor (CDKN2A) leading to an increase in proliferation* ³³. *In addition, other study has shown that PDGF, highly expressed in DIPGs, has the ability to stabilize E2F1* ³⁴.”

In 699. Should be 15 days post-cell implantation. We have changed this and highlighted in red. After the revision this change is located in page 32, line 710.

Fig 3 A-D, H, I. What does the blue shaded area represent? The shaded area represent the 50-day interval after cell implantation. Since the different models have different kinetics we highlighted the first 50 days of the experiment in order to provide a better perspective of the time frame of the survival experiment and how they related between the different models. We have included an explanation of what the shaded area represents in the Figure 3 legend page 32, lines 696-697 and 710-711: “*The shaded area represents a 50-day interval from the time of cell implantation*”.

REVIEWERS' COMMENTS:

Reviewer #1 (Remarks to the Author):

The authors have satisfied all questions and ambiguities.
I have no further comments.

Reviewer #2 (Remarks to the Author):

Authors have satisfactorily responded to comments.